# MultiSite Assembly of Gateway Induced Clones (MAGIC): a flexible cloning toolbox for use in vertebrate model systems

**William B. Gillespie, III[1], Yuwen Zhang[1], Oscar E. Ruiz[2], Juan Cerda, III[2], Joshua Ortiz-Guzman[3,4,5], Michelle Sherman[1], Williamson D. Turner[2], Gabrielle Largoza[1,2], Lili E. Mosser[1], Esther Fujimoto[6], Chi-Bin Chien[6], Kristen M. Kwan[6], Benjamin R. Arenkiel[3,4,5], W. Patrick Devine[7] and Joshua D. Wythe[1,2,8,9,10,11,*]**

## ABSTRACT

Here, we present MultiSite Assembly of Gateway Induced Clones (MAGIC), which leverages Gateway-based recombinatorial cloning technology for rapid, modular assembly of plasmids to facilitate transgenesis in cells and vertebrate animal models. The MAGIC collection of plasmids spans a range of *in vitro* and *in vivo* uses, from tools for optically and chemically tunable gene expression, to simultaneous expression of microRNAs and fluorescent reporters, to a suite of distinct subcellular compartmental fluorescent reporters, to Cre and Dre recombinase-dependent gene expression. MAGIC system components are compatible with existing MultiSite Gateway Tol2 systems currently used in zebrafish and mammalian lentiviral and adenoviral Destination vectors, allowing rapid cross-species experimentation. The kit also includes novel vectors for stable transgene integration into host genomes *in vitro* or *in vivo* when used with *piggyBac* transposase, I-Sce meganuclease or Tol2 transposase. Collectively, the MAGIC system facilitates transgenesis in cultured mammalian cells, mouse, chick and zebrafish embryos, enabling the rapid generation of innovative DNA constructs for biological research due to a shared, common plasmid platform.

KEY WORDS: *piggyBac*, Gateway, Cloning, Fluorescence, Dre/*rox*, Cre/*lox*, Tet/dox, LexA/LexOp

## INTRODUCTION

From genome editing to driving expression of fluorescent reporters for live imaging of cellular dynamics, transgenesis is an essential part of modern experimental biology. A successful blueprint for flexible and rapid recombinant DNA cloning and transgenesis was created by the zebrafish community when they combined Tol2 transposon-mediated genomic integration (Kawakami et al., 2000) with recombination-based technologies using MultiSite Gateway cloning (Cheo et al., 2004; Hartley et al., 2000).

These and other Gateway compatible Tol2 'toolkits' facilitate the modular assembly of various DNA elements, including a promoter, fluorescent reporters, cDNAs, and more (Campeau et al., 2009; Fowler et al., 2016; Kwan et al., 2007; Villefranc et al., 2007). Additionally, standalone Gateway-compatible transgenic expression plasmids (Campbell et al., 2012; Sigl et al., 2014; White et al., 2011) and open reading frame libraries (so called ORFeomes) from humans (Bechtel et al., 2007; Lamesch et al., 2007, 2004; Reboul et al., 2003; Rual et al., 2004; Yang et al., 2011), worms (Dupuy et al., 2004; Lamesch et al., 2004; Reboul et al., 2003), frogs (Grant et al., 2015) and bacteria (Brandner et al., 2008; Dricot et al., 2004; Maier et al., 2012; Rajagopala et al., 2010) extended the utility of this system beyond zebrafish. However, many of these other toolkits do not employ the same arrangement of MultiSite Gateway recognition sequences, limiting the utility of these tools across experimental platforms.

For this reason, we present MAGIC, a collection of Entry and Destination vectors, generated *de novo* or derived from existing plasmids, to facilitate transgenesis in mammalian systems. Entry vectors in the kit range from tissue-specific promoters, to tunable/inducible gene expression systems, to fluorescent reporters, site-specific recombinase (SSR) vectors (Cre/*loxP* and Dre/*rox*) bicistronic expression systems, dual fluorescence and microRNA expression plasmids, optogenetics, and beyond. We also supply several mammalian-compatible Destination plasmids for *piggyBac* transgenesis, lentiviral and adeno-associated viral transduction, gene targeting and constitutive expression in mammalian cells. All Entry vectors in the kit are compatible with existing zebrafish Tol2 Gateway kits, and we also include a handful of novel Tol2 Destination vectors for temporal and tissue-specific transgenesis in zebrafish. Finally, we offer a relational, open-source database to facilitate record keeping of these and other plasmids and related information (sequences, files, etc.).

[1]Department of Cell Biology, University of Virginia School of Medicine, Charlottesville, VA 22903, USA. [2]Department of Integrative Physiology, Baylor College of Medicine, Houston, TX 77030, USA. [3]Department of Molecular and Human Genetics, Baylor College of Medicine, Houston, TX 77030, USA. [4]Jan and Dan Duncan Neurological Research Institute, Texas Children's Hospital, Houston, TX 77030, USA. [5]Department of Neuroscience, Baylor College of Medicine, Houston, TX 77030, USA. [6]Department of Human Genetics, University of Utah, Salt Lake City, UT 84112, USA. [7]Department of Pathology, University of California San Francisco, San Francisco, CA 94115, USA. [8]Department of Neuroscience, University of Virginia School of Medicine, Charlottesville, VA 22903, USA. [9]Brain, Immunology, and Glia (BIG) Center, University of Virginia School of Medicine, Charlottesville, VA 22903, USA. [10]Robert M. Berne Cardiovascular Research Center, University of Virginia School of Medicine, Charlottesville, VA 22903, USA. [11]University of Virginia Comprehensive Cancer Center, University of Virginia School of Medicine, Charlottesville, VA 22903, USA.

*Author for correspondence ( jwythe@virginia.edu)

W.P.D., 0000-0003-4634-8830; J.D.W., 0000-0002-3225-2937

## RESULTS
### Overview

Gateway recombination-based cloning enables the rapid and robust insertion of DNA fragments from so-called 'Entry clones' into plasmids known as 'Destination' vectors (Fig. S1). Gateway cloning has been successfully exploited to facilitate transgenesis in some species, such as zebrafish, by merging it with robust transgenic technologies, including *Tol2* transposase (Kwan et al., 2007;

Villefranc et al., 2007) *Tol1* transposase (Shin et al., 2016) and Phi31 integrase (Lalonde et al., 2024; Mosimann et al., 2013). Furthermore, the constant addition of novel Entry clones ensured the utility of these approaches (Don et al., 2017; Fowler et al., 2016; Kemmler et al., 2023). In zebrafish MultiSite Gateway transgenesis kits, there are three varieties of Entry clones: 5′ Entry clones (p5E), middle Entry clones (pME) and 3′ Entry clones (p3E). p5E vectors provide a promoter, while pME vectors contain inserts, such as a fluorescent reporter or gene of interest. The p3E vectors usually contain a polyA signal, a fluorescent reporter (e.g. IRES-mCherry), or an epitope tag (e.g. FLAG). MultiSite Gateway cloning enables the rapid assembly of these three modules ([promoter] [coding sequence] [tag or reporter]) into a single Destination plasmid (Fig. S2). However, few MultiSite mammalian expression vectors follow the same modular MultiSite Gateway recombination logic as the zebrafish Tol2, Tol1 and Phi31 systems, limiting the utility of these collections to other model systems.

Herein, we describe a mammalian-compatible Gateway toolkit with several tissue-specific promoters, more than 22 unique fluorescent reporters, as well as *piggyBac* transposase-compatible MultiSite Destination plasmids to facilitate stable transgenesis in amniote model systems. We also provide Cre and Dre recombinase-dependent Destination vectors for gene targeting in mice, and middle Entry compatible lentiviral and AAV Destination plasmids, as well as novel *Tol2*-based backbones for temporal and tissue-specific gene expression studies in zebrafish.

## 5′ Entry clones
Sequences used for BP recombination sites are listed in Table S1, and a complete list of p5Es can be found in Table S2.

### 5′ Entry promoters for broad expression
Four p5E promoter constructs are included for broad, constitutive expression in mammalian cells at varying levels (Qin et al., 2010). p5E-CAG (or CAGGS) drives strong, ubiquitous expression in mammalian cells (Miyazaki et al., 1989; Niwa et al., 1991). p5E-EF-1α, which contains the human elongation factor 1α (EF-1α; *EEF1A1*) promoter and intron, is also a robust promoter, while p5E-EFS contains only the EF-1α core promoter (known as EF-1α short, or EFS) for modest expression (Schambach et al., 2006). For moderate expression in mammalian systems, we also provide p5E-*Ubi*, which contains the promoter from the human ubiquitin C gene, *UBC* (Schorpp et al., 1996).

### 5′ entry promoters for chemically inducible gene expression
In addition to constitutively active mammalian promoters, we also provide inducible promoters. The Tet system enables inducible gene expression in mammalian cells (Gossen and Bujard, 1992; Gossen et al., 1995). We created two Gateway-compatible, tetracycline-regulatable promoter variants. The first, p5E-TetO$_{8X}$-CMV$_{min}$, contains a second-generation Tet response element (TRE) composed of eight *tet* operator sequences located upstream of a minimal CMV promoter (Agha-Mohammadi et al., 2004; Kim et al., 2016b). The second, p5E-TRE-3GV, contains a third-generation TRE promoter composed of seven *tet* operator sites upstream of a pTight hybrid sequence (Clontech, 2003) followed by a minimal CMV promoter (Lagrange et al., 1998) to ensure low basal activity and maximal responsiveness to doxycycline (Loew et al., 2010).

We also provide p5E-lexAOP-c-Fos for mifepristone-inducible gene expression in zebrafish and mammalian cells (Emelyanov and Parinov, 2008; Nguyen et al., 2012), as described further in the 'LexA/LexO tools' section.

### 5′ Entry promoters for optically inducible gene expression
Woo and colleagues created an optogenetic system for regulating gene expression in zebrafish (Reade et al., 2017). The light-responsive prokaryotic transcription factor EL222 consists of a light-oxygen-voltage (LOV) domain followed by a helix-turn-helix (HTH) DNA-binding domain (DBD) (Motta-Mena et al., 2014). When illuminated with blue light (450-490 nm), EL222 dimerizes and binds to a regulatory element known as C120 (Nash et al., 2011; Rivera-Cancel et al., 2012). Combining the transactivation domain of KalTA4 with EL222 generates TAEL (TA4-EL222), a photo-activatable gene expression system with rapid on/off kinetics (Reade et al., 2017). Placing a murine *c-Fos* basal promoter downstream of the C120 transcriptional cassette enabled high expression with minimal background activity (Dorsky et al., 2002; LaBelle et al., 2021; Scott and Baier, 2009). We modified this design by adding an SV40 polyA followed by a transcriptional pause site, then a NOS terminator, upstream of the 5× C120 sites followed by the *c-Fos* minimal promoter (p5E-*5×C120-c-Fos-minpro*) to create TAEL-responsive vectors for optical control of gene expression.

### 5′ Entry promoters for tissue-specific expression
The MAGIC toolkit contains multiple pan-endothelial specific promoters, specifically murine VE-cadherin (*Cdh5*) and human *ICAM2* (both the full-length and minimal promoters) (Cowan et al., 1998; Dai et al., 2004; Huang et al., 2017; Wang et al., 2016), human claudin 5 (*CLDN5*) (de Leeuw et al., 2014), and an enhancer of murine delta-like 4 (*Dll4*) that drives arterial endothelial expression in mice and zebrafish (Wythe et al., 2013).

We provide cardiac-specific promoters for use in zebrafish and mice: the zebrafish *myl7* promoter (Kwan et al., 2007) and the human cardiac troponin T (*TNNT2*) promoter (Werfel et al., 2014). The kit also contains a pan-muscle promoter (-*503unc*) (Berger and Currie, 2013) and a zebrafish-specific neural crest driver (*crestin*) (Kaufman et al., 2016) (Table S2).

For studies in mice, chick, and cell culture models, we generated a p5E human *GLAST* (*EAAT1*, *SLC1A3*) promoter clone to drive expression in radial glia and astrocytes (Chen and LoTurco, 2012; Kim et al., 2003).

Finally, we also provide p5E-MCS-*c-Fos-β-globin*, where an extensive multiple cloning site (MCS) allows for insertion of any enhancer element upstream of a minimal murine *c-Fos* promoter followed by the rabbit *beta-globin* intron for use in zebrafish.

Collectively, this suite of varying strength mammalian promoters, as well as inducible and tissue-specific promoters, will facilitate experiments in eukaryotic cell lines, as well as in vertebrate animal models.

## Middle Entry clones
The MAGIC system uses middle Entry clones (pMEs) to supply fluorescent reporters or genes of interest. A complete list of all pME vectors is provided in Table S3.

### Fluorescent reporters
The MAGIC kit includes pMEs encoding 22 distinct fluorescent proteins (FPs) spanning the imaging spectrum, from blue [mTagBFP2 (Subach et al., 2011)] to cyan [AmCyan1 (Matz et al., 1999; Richards et al., 2002); mCerulean (Rizzo et al., 2004); mCerulean3 (Markwardt et al., 2011); mTurquoise2 (Goedhart et al., 2012)], green [EGFP (Cormack et al., 1996), mClover3, mNeonGreen], red [mCherry (Shaner et al., 2004); mRuby2 (Lam et al., 2012); mKate2 (Shcherbo et al., 2009); mScarlet-I (Bindels et al., 2017)] and far red [mCardinal (Chu et al., 2014)]. Newer reporters, such as the heavily mutated,

cysteine-free mScarlet3 and its fast-maturing derivative mScarlet-I3 (Gadella et al., 2023), and stable derivatives mScarlet-3H (mYongHong/mScarlet3-M163H/mScarlet3-H) (Xiong et al., 2025; Xu et al., 2024 preprint) and mScarlet3-S2 (M163H, M66Q) (Xu et al., 2024 preprint), as well as the stable green fluorescent protein variant StayGold [(n2)oxStayGold(C4)] (Hirano et al., 2022) and its monomeric derivatives mStayGold (Ando et al., 2024) and mBaoJin (Zhang et al., 2024), were included for their reported superior stability and brightness. Near infrared (iRFP) and tandem iRFP dimer (tdiRFP) were chosen because their wavelength is outside the absorption spectrum of hemoglobin and melanin (Zhang et al., 2012). While bacterial phytochrome-derived near infrared FPs, iRFP and tdiRFP (iRFP670) (excitation 643 nm/emission 680 nm) require an intermediate of heme metabolism, biliverdin, to form a chromophore (Auldridge and Forest, 2011; Shcherbakova and Verkhusha, 2013), they produce a robust signal in zebrafish with no supplementation (Cook et al., 2019).

Permutations of these various FPs are included to label different subcellular compartments. For nuclear labeling, we provide pME plasmids containing a nuclear localization signal (NLS) fused to an FP followed by a stabilizing woodchuck hepatitis virus post-transcriptional regulatory element (WPRE) and bGH polyA (nls-mTagBFP2-FLAG, nls-mKate-V5, nls-EGFP), as well as histone H2B or H2A chimeric FPs [pME-H2B-mCerulean; pME-mCardinal-H2B; pME-H2B-mStayGold-WPRE; pME-H2B-V5-(n2)oxStaGold(c4); pME-H2B-mBaoJin-WPRE; pME-H2B-mScarlet3-H; pME-H2B-mScarlet3-S2], and an anti-lamin B nanobody fused to mNeonGreen to label the nuclear lamina (pME anti-laminB-mNeonGreen), as well as a fusion of the nuclear envelope protein Sun1 to two copies of superfolder (sf)GFP (pME-Sun1-2×sfGFP-6×Myc).

For visualizing cytoskeletal dynamics, we provide multiple FPs conjugated to Lifeact, which recognizes filamentous actin (F-actin) in eukaryotic cells (pME-Lifeact-mRuby2-3×MYC, pME-Lifeact-mScarlet, pME-Lifeact-mCherry, pME-Lifeact-EGFP and pME-MCS-P2A-Lifeact-mRuyb2-3×MYC). Unlike fluorescent phalloidin dye, chimeric actin-FP fusion proteins, or bulky anti-actin antibodies, the small size of Lifeact (17 amino acids) enables the live study of endogenous intracellular actin (Riedl et al., 2008). Because artifacts may result from overexpression of Lifeact reporters (Flores et al., 2019; Spracklen et al., 2014; Xu and Du, 2021), we also provide an α-actin nanobody (VHH) fused to sfGFP, mCherry and the versatile HaloTag (Grimm et al., 2017; Los et al., 2008).

We also created general cytoplasmic fluorescent reporters (pME-AmCyan1, pME-V5-mScarlet-I), and FP chimeric fusions to human giantin (golgin B1) to label the Golgi [pME-mScarlet3-Giantin, pME-(n2)oxStayGold(c4)-Giantin], as well as tubulin-FPs to visualize cytoskeletal dynamics (pME-mCerulean3-TUB).

A handful of clones encode multiple FPs to enable simultaneous labeling of different subcellular structures. For example, pME-Vhh-actin-sfGFP-P2A-iRFP-caax allows for sfGFP labeling of actin and iRFP labeling of the plasma membrane in the same cell. Recombination of these Entry clones into an expression-ready mammalian Destination vector and transient transfection followed by confocal microscopy confirmed the fidelity of these Entry clones (Fig. 1, Fig. S3).

## Cell cycle tools

Levels of Cdt1 and its inhibitor geminin oscillate in a mutually exclusive manner during the cell cycle. Fluorescence Ubiquitin Cell Cycle Indicator (FUCCI) (Sakaue-Sawano et al., 2008) leverages this degradation system to report on G1 and S/G2/M state via differential accumulation of Cdt1 and geminin fluorescently tagged

fusion proteins (Fig. 2A). In the FastFUCCI reporter (Koh et al., 2017), a T2A peptide cleavage sequence ensures equimolar expression of monomeric Kusabira Orange 2 (mKO2) fused to human CDT1 (amino acids 30-120) [mKO2-hCDT1(aa 30-120)] and monomeric Azami Green (mAG) fused to human geminin (amino acids 1-110) [mAG-hGem(1-110)]. We generated pME-FastFUCCI (Fig. 2B) and pME-FastFucci-P2A-H2A-tdiRFP to label all nuclei, regardless of cell cycle state, with a tdiRFP reporter. Generation of pCAGEN-FastFUCCI, pCAGEN-pCAGEN-FastFUCCI-IRES-myr-mTagBFP and pB-EF1a-FastFUCCI-IRES-3×NLS-mTagBFP2, followed by transient transfection of HEK-293T cells and confocal imaging, confirmed that combining FastFUCCI with a cell membrane- or nuclear-localized fluorescent reporter enables tracking all transfected cells, regardless of their cell cycle state (Fig. 2C,D, Fig. S4).

We also provide a zebrafish-compatible FUCCI (Bouldin et al., 2014; Jerafi-Vider et al., 2021) in a pME, pME-zFUCCI (Fig. 2E), as well as pME-zFUCCI-P2A-H2A-tdiRFP, which should label every cell with H2A.Z-tdiRFP. Injection of a *Tol2*-flanked vector with a ubiquitin promoter driving zFUCCI followed by confocal microscopy confirmed the functionality of this tool (Fig. 2F,G).

## microRNA tools

For studying microRNAs, we provide an Entry vector in which a multiple cloning site is embedded within an artificial intron separating the 5′ and 3′ halves of EGFP (pME-EGFP-miR-MCS) to enable the simultaneous transcription of EGFP and a primary microRNA (pri-miRNA) (Bartel, 2004). Herein, we provide pME-EGFP-hs_*miR-126*, which contains the endothelial-enriched miRNA *miR-126* embedded within the human epidermal growth factor like-domain 7 (*EGFL7*) gene, which we previously validated in zebrafish and mammalian cell culture models (Fish et al., 2008). We also provide pME-EGFP-mm_*miR-500*, which contains *miR-500* embedded within an intron of the murine chloride voltage-gated channel 5 (*Clcn5*) gene (Wheeler et al., 2006). The MAGIC kit includes EGFP-miR-MCS, *miR-126* and *miR-500* expression vectors for use in cultured mammalian cells.

## Cre and Dre SSR tools

Herein, we also provide pMEs for use with SSRs. The SSR Dre exhibits robust activity *in vitro* and *in vivo* in mice, with minimal cross-recognition of *loxP* sites (Anastassiadis et al., 2009; Devine et al., 2014). We generated pME-Dre-NLS and pME-DreERT2 for tamoxifen-inducible Dre activity, and include CAG-driven expression vectors for both variants (pCAGEN-Dre-nls, pCAGEN-DreERT2).

A suite of plasmids encoding FP-Cre fusions are also included, such as pME-nls-EGFP-Cre and pME-nlsTagBFP-Cre, as well as bi-cistronic FP-improved Cre (iCre) (Kaczmarczyk and Green, 2001) plasmids (pME-mTurquoise-P2A-iCre, pME-EGFP-P2A-iCre, pME-mScarlet-I-P2A-iCre). We also provide pME-se-iCreI, in which a synthetic chimeric intron separates the 5′ and 3′ halves of the iCre open reading frame to prevent recombination in prokaryotic systems, and the entire expression cassette is flanked by *loxP* sites to create a self-excising iCre insert. pME-loxP-3×STOP-*loxP*, which can act as a transcriptional stopper when placed upstream of a p3E fluorescent reporter (see below), as well as pME-*loxP*-3×NLS-mCherry-stop-V5-SV40pA *loxP* are provided to generate Cre-dependent reporter vectors.

pME-LiCre contains a single chain, fast-responding, light-activatable Cre (LiCre). Developed by the Yvert lab (Duplus-Bottin et al., 2021), LiCre is solely activated by illumination without additional chemicals or co-factors. We generated pCAGEN-LiCre and validated its activity *in vitro* (Fig. 3A-C). Notably, lentiviral

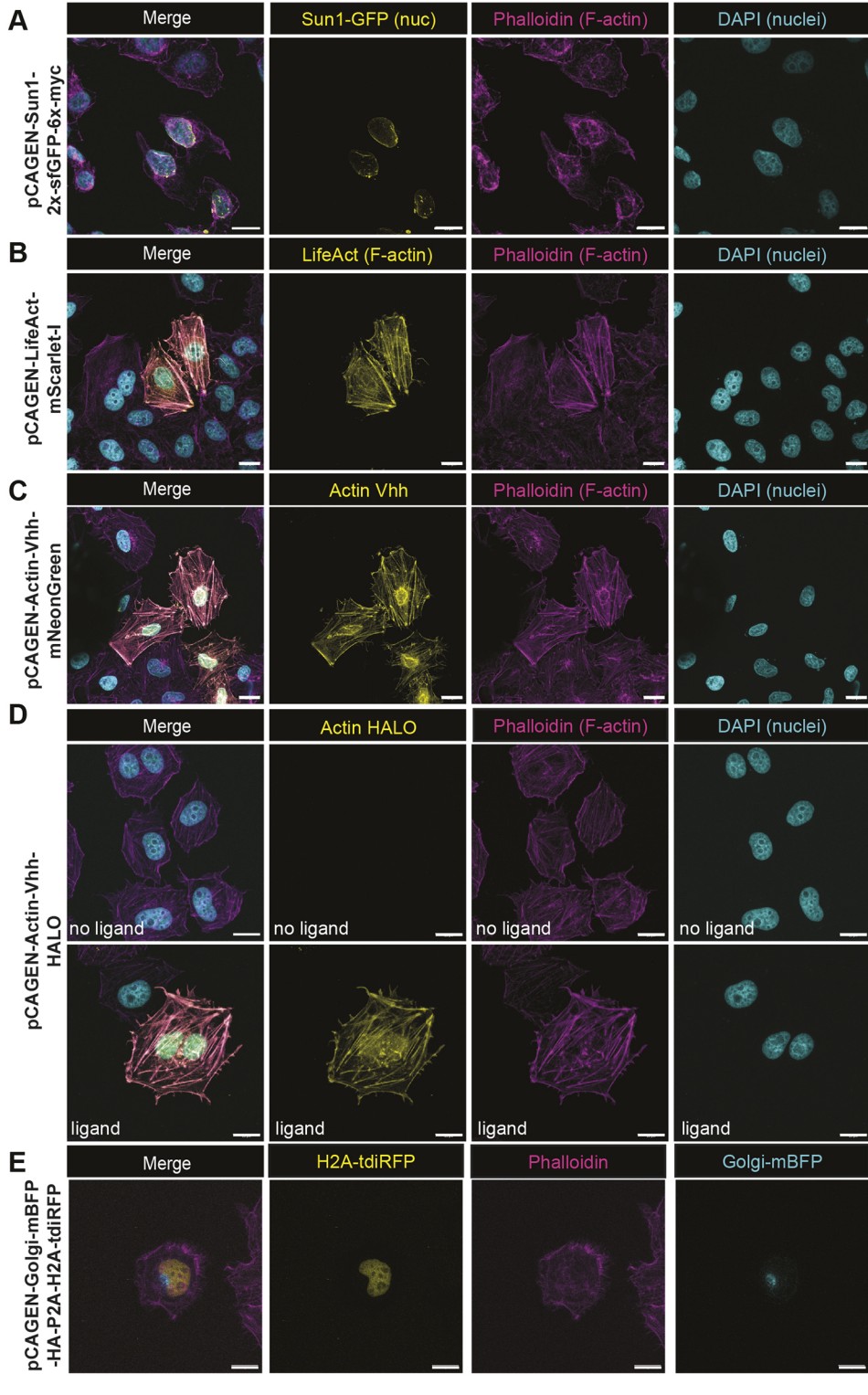

**Fig. 1. Subcellular labeling using MAGIC kit components.** Clones were made using middle Entry clones (pMEs), described in Table S3, and the pCAGEN-DEST vector to facilitate labeling of (A) nuclei with a 2× superfolder GFP fusion to the nuclear envelope protein Sun1, (B) filamentous actin with a fusion of mScarlet to the short F-actin probe Lifeact, (C) a pan-actin probe generated from a fusion of a nanobody that recognizes actin (actin Vhh) fused to mNeonGreen for a live reporter, as well as (D) a fusion of the same actin nanobody fused to Halo tag for flexible use of antibody or dye labeling (in this case Janelia dye JFX646), with and without dye, and (E) dual labeling of the Golgi and nucleus with GalT-mTagBFP and an H2A-tdiRFP probe. All pME constructs were LR recombined into pCAGEN and transiently transfected into HeLa cells. Some cells were fixed, permeabilized and counterstained with DAPI to label nuclei and phalloidin to label F-actin, or stained with Hoechst while live, and then imaged by confocal microscopy. Scale bars: 20 µm.

transduction of pLenti-CMV-LiCre via stereotactic injection and optical induction 14 days later led to robust recombination of a Cre-dependent fluorescent reporter in mice (Fig. 3D-G).

**Light-inducible transcriptional activators**

We created pME-TAELn-pause-5×C120, an 'all in one' plasmid that encodes the chimeric blue light (450 nm)-inducible transcriptional activator TAEL2.0 (LaBelle et al., 2021; Reade et al., 2017) followed by a SV40 polyA signal and a transcriptional pause site to prevent errant read-through from p5E supplied promoters, then a NOS terminator sequence, and finally an effector cassette composed of five C120-binding domains and a minimal murine *c-Fos* promoter (Fig. 3H). Injection of pTol2-Ubi-TAEL2.0-C120-mCherry into *Tg(kdrl: EGFP)* embryos confirmed that mCherry expression was induced only in animals exposed to blue light, validating the use of this all-in-one, optogenetically regulated gene expression system (Fig. 3H-J).

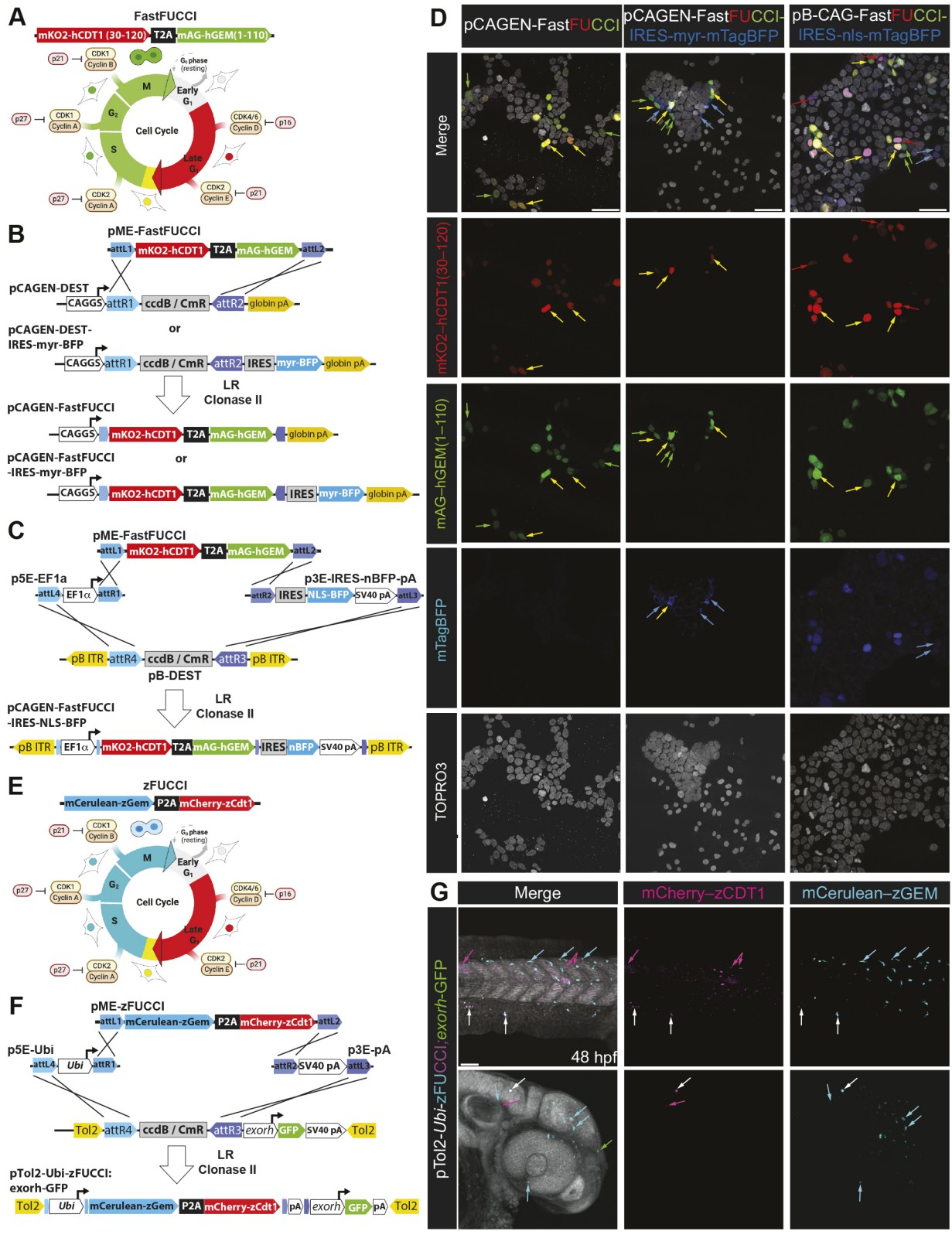

**Fig. 2.** See next page for legend.

## LexA/LexO tools

The hormone inducible Lex transactivator/Lex operon (LexA/LexO) system is a powerful tool for inducible gene expression (Emelyanov and Parinov, 2008; Nguyen et al., 2012). In the presence of the progesterone agonist mifepristone (RU-486), the chimeric LexPR transactivator binds to the synthetic LexA operon (LexOP) to drive expression of an effector cassette.

The two elements of the LexA/LexO system are typically combined in trans (Emelyanov and Parinov, 2008) (Fig. 4A,B). We tested whether the transcriptional activator and regulatory

**Fig. 2. Labeling cell cycle dynamics using FUCCI.** (A) Schematic of the FastFUCCI system originated by Koh and colleagues (Koh et al., 2017) in which fusion of mKO2 to human CDT1 (amino acids 30-120) followed by a T2A peptide and mAG fused to human geminin (amino acids 1-110) enables fluorescent visualization of progression through the mammalian cell cycle. (B) Diagram of pME FastFUCCI and LR recombination into pCAGEN-DEST and pCAGEN-DEST-IRES-myr-BFP. (C) MultiSite recombination of p5E-EF1a, pME-FastFUCCI and p3E-IRES-nls-BFP into pB-DEST. (D) Transient transfection into HEK-293T cells confirms activity of the reporter, and compatibility with myr- and nls-BFP reporters. Red arrows denote cells in late G1, while yellow denotes cells transitioning from late G1 to S phase, and green denotes cells in S-G2-M phase. Blue denotes cells expressing the myr- or nls-BFP reporter. Green arrows indicate mAG-hGEM (S-G2-M) phase cells, yellow indicates cells transitioning from late G1 to S, red indicates late G1 cells, while blue indicates the IRES-myr-BFP reporter and labels transfected cells. Images in the top row ('Merge') are also shown in Fig. S4. (E) Schematic of the zFUCCI system from Kimelman and colleagues (Bouldin et al., 2014). (F) MultiSite recombination of p5E-*Ubi*, pME-zFUCCI and p3E-pA into pTol2-DEST-exorh-GFP. (G) Tol2-mediated transient transgenesis followed by confocal imaging at 48 hpf confirm the functionality of pME-zFUCCI, with clones in S/G2/M phase colored in cyan, late G1 phase cells in magenta, overlapping cells in white, and the exorh-GFP reporter indicated in green. Aqua-colored arrows indicate mCerulean-positive cells (S-G2-M), white indicates late G1/S transitioning cells, magenta indicates late G1, and green indicates exorh::GFP-positive cells. Scale bars: 50 μm.

elements could be combined to create a single vector system. However, a drawback of the LexA/LexO chemically inducible gene expression system in zebrafish is the reportedly slow off-rate kinetics (Emelyanov and Parinov, 2008). Accordingly, we compared three pME LexA/LexO variants (Fig. 4C). The first-generation LexA/LexO clone contains a chimeric LexPR transactivator (LexA) and a LexOP (LexO) consisting of a SV40 polyA followed by a terminator sequence from the *Agrobacterium tumefaciens* nopaline synthase (NOS) gene, and four ColE1 sites and a 35S cauliflower mosaic virus minimal promoter (CaMV 35S promoter) (Zuo et al., 2000). In the second-generation construct, pME-LexPR-mODC, we fused the LexPR transactivator to the PEST domain of murine ornithine decarboxylase (mODC) to reduce the protein half-life (Li et al., 1998; Loetscher et al., 1991). The third-generation clone retained this destabilized LexPR and altered the LexOP, exchanging the NOS terminator for a RNA pol II transcriptional pause site, increasing the number of ColE1 lexA operator sites from four to six, and replacing the CaMV 35S promoter with the mouse minimal *c-Fos* promoter.

To determine whether an 'all in one' LexA/LexO vector system could control gene expression, and ascertain the impact of these modifications, we tested them in zebrafish (Fig. 4D,E). The first-generation LexA/LexO construct induced robust nls-GFP expression, even in the absence of RU486 treatment. Destabilizing the LexPR had minimal impact, as embryos also displayed nls-EGFP signal in the absence of RU486. However, the third-generation construct demonstrated that an all-in-one approach can be used for RU486-inducible gene expression in zebrafish (Fig. 4F-H). For cases in which a dual-construct approach is needed, we provide the third-generation LexOP as a standalone p5E vector.

### 3′ Entry clones
3′ Entry clones are designated as p3E, and the *attR2/L3* site flanked DNA inserts usually supply a polyadenylation (polyA) signal, an internal ribosomal entry site (IRES) and bicistronic fluorescent reporter, or create a direct fusion to a FP. Table S4 lists all p3Es provided in the MAGIC kit.

### polyA signals
The SV40 late polyA stabilizes mRNA transcripts and promotes translation in multiple species (Carswell and Alwine, 1989; Kwan et al., 2007; Matsumoto et al., 1998). As p3E SV40 late polyA vectors are available at Addgene (Addgene plasmid #49004 and Addgene plasmid #75174), we generated rat EF1a polyA. For enhanced transcript stability and reduced transgene silencing, we provide p3E-WPRE-SV40-pA, which contains the 588 bp WPRE (Donello et al., 1998; Paterna et al., 2000). Using stronger or weaker polyA signals (Peterman et al., 2024), as well as the inclusion of an intron in the promoter (Powell et al., 2015; Ramezani et al., 2000) may negate the impact of including a WPRE on transcript stability and increased translation. Thus, we also include p3E-WPRE-bGH-pA, which contains the weaker bovine growth hormone (bGH) polyA.

### Fluorescent reporters and gain-of-function tools
We provide a suite of p3E vectors to insert downstream of pME *rox*- or *loxP*-flanked stop cassettes or 3′ to a transcriptional effector, such as pME-5×C120 TAEL, including p3E-mCherry-SV40pA, p3E-H2A-mCherry-SV40-pA, p3E-V5-mScarlet-SV40-pA, p3E-Actin-Vhh-mNeonGreen-HA and p3E-mTagBFP2. We also provide p3E-IRES-reporters, including p3E-IRES-3×FLAG-3×NLS-mScarlet-I3-SV40-pA, p3E-IRES-3×FLAG-3×NLS-mTagBFP2-SV40-pA, p3E-IRES-V5-(n2)oxStayGold(c4)-pA, p3E-IRES-H2A-mCherry-SV40-pA and p3E-IRES-Luciferase.

### Tet-inducible transactivators
To create 'all in one' Tet-inducible vectors, we generated a p3E plasmid containing a bGH polyA, followed by a synthetic polyA and a transcriptional pause from the human α2 globin gene (*HBA2*) to prevent Pol II read-through, then the minimal human *EFS* promoter driving expression of a third-generation reverse tetracycline transactivator (rtTA-3G; also known as Tet-On 3G, or rtTA-V10; Zhou et al., 2006). In this Tet-On system, doxycycline induces a conformational change that allows the transactivator protein to bind *tet* operator sequences within a TRE to drive expression of an effector cassette. Thus, one can assemble a p5E-TRE promoter (we supply two variants in the kit), followed by a pME containing a gene of interest, and p3E-bGH-pA-EFS-rtTA-3G to create an 'all in one' tetracycline-responsive expression vector, rather than employing a binary, two-vector system.

### Destination vectors
A complete list of single and MultiSite Destination vectors can be found in Table S5.

Several MultiSite Gateway-compatible Destination vectors for transgenesis in zebrafish have been described, including *Tol2* transposase compatible kits (Kemmler et al., 2023; Kwan et al., 2007; Villefranc et al., 2007), an analogous *Tol1* transposase system (Shin et al., 2016), and a recently published attP/Phi31-targeted integration platform (Lalonde et al., 2024). While these collections are optimized for use in teleosts, the MAGIC toolkit predominantly contains mammalian-compatible Destination vectors for MultiSite cloning, as well as Cre/*lox* and Dre/*rox* single-site Destination vectors to facilitate insertion of pMEs into murine gene targeting vectors. We also provide a suite of single site, 'plug and play' Destination vectors with ubiquitous or tissue-specific promoters for *in vitro* and *in vivo* studies. However, we have also optimized *Tol2* MultiSite Destination vectors to enable tissue-specific, heat shock-inducible gene expression in zebrafish. A complete list of all expression vectors generated for this study and included in the MAGIC kit be found in Table S6.

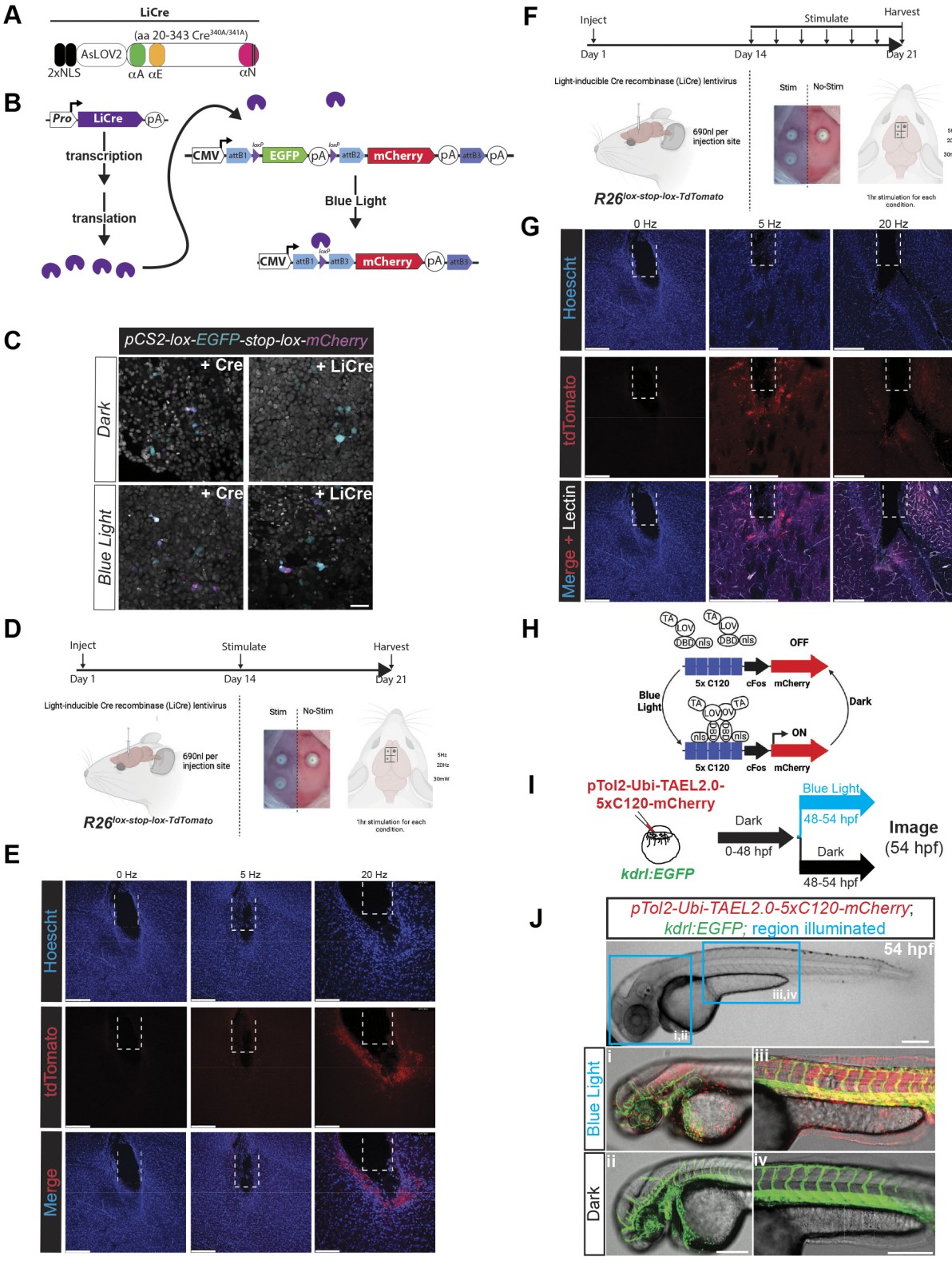

**Fig. 3.** See next page for legend.

## CAGGS Destination vectors for single Entry cloning

Multiple pME-compatible Destination plasmids (e.g. in which a mammalian promoter is upstream of an *attR1-ccd*B-CmR-*attR2* Gateway recombination cassette) exist for transient, ubiquitous expression in mammalian cells (Fig. 5A). A particularly versatile

group of Destination vectors utilize the pCS2 backbone (Kwan et al., 2007; Turner and Weintraub, 1994; Villefranc et al., 2007). However, the CMV-IE enhancer/promoter cassette in pCS2 is often less active than synthetic alternatives. One prominent example is CAG (Niwa et al., 1991), a fusion between the CMV

**Fig. 3. Optogenetic control of gene expression tools.** (A) Schematic of the LiCre cassette, composed of a 2× NLS tag followed by the photoreceptor domain of *Avena sativa* (AsLOV2) fused to a N-terminally truncated variant of Cre recombinase with destabilizing mutations in its C-terminal domains. (B) LiCre activity is induced within minutes of illumination with blue light without the need for additional chemicals or components. (C) Transfection of LiCre in mammalian cells, along with a Cre-dependent switch reporter, confirms activation in the presence, but not absence, of blue light. (D) Experimental design for lentiviral transduction by stereotactic injection of the murine brain, followed 2 weeks later by illumination. (E) Robust recombination of a *Rosa26^lox-stop-lox-TdTomato* fluorescent reporter, as detected by TdTomato fluorescence within the murine brain, is evident 1 week after a single hour of stimulation at 5 Hz and 20 Hz. Dashed lines indicate fiber-optic implant placement. (F) Experimental design for repeated illumination over the course of 1 week following viral transduction. (G) Robust recombination is evident following serial illumination, 1 h/day at 5 Hz. Variable Cre activity, as evidenced by TdTomato. Dashed lines indicate fiber-optic implant placement. (H) Design of the TAEL2.0 system, pioneered by Woo and colleagues (LaBelle et al., 2021), where fusion of the KalTA4 transcriptional activator (TA) domain to blue light-oxygen-voltage-sensing (LOV) domain and DNA-binding domain (DBD) of the *Erythrobacter litoralis* HTCC2594 transcription factor EL222, followed by a C-terminal NLS tag, allows for 450 nm light-induced conformational change and dimerization of TAEL (KalTA4-EL22), allowing it to bind and initiate transcription from the C120 regulatory element upstream of a minimal *c-Fos* promoter and mCherry reporter cassette. (I) Experimental paradigm for injection into a endothelial EGFP reporter, *Tg(kdrl:EGFP)* at the one-cell stage, followed by illumination with blue light. (J) Whole-mount, sagittal view, phase image of a 54 hpf embryo containing the *Tg(kdrl:EGFP)* reporter and illuminated in the indicated, blue boxed regions (i-iv) in the upper panel. Blue light illumination led to robust induction in pTol2-Ubi-TAEL-N-5×C120-mCherry-injected embryos within a few hours of exposure to blue light, while embryos maintained in the dark did not show induction of mCherry. Scale bars: 50 µm (C); 250 µm (E,G,J).

immediate-early enhancer and the *AG* promoter [a hybrid between the promoter and first intron of the chicken β-actin gene and a fragment spanning intron 2 and exon 3 of the rabbit *beta-globin* gene that contains an effective splice acceptor (Keller, 1984; Miyazaki et al., 1989)]. Unlike pCS2, which contains SV40 late polyadenylation signal sequence, pCAGGS employs a rabbit *beta-globin* polyA signal sequence, and an SV40 origin of replication. We converted pCAGEN, a pCAGGS derivative with an expanded multiple cloning site (Matsuda and Cepko, 2004), into pCAGEN-DEST (Fig. 5B).

To compare pCAGEN to existing mammalian DEST expression vectors, a pME with a consensus Kozak sequence preceding luciferase, followed by a P2A cleavage peptide, then an H2A-mCherry fusion protein was LR recombined into pCS2-Dest and pCAGEN-Dest (Fig. 5B). Luciferase assays and western blotting following transient transfection of HEK-293T cells confirmed increased activity of pCAGEN compared to pCS2 (Fig. 5C,D).

Because stabilizing mRNA production may be useful in some contexts, we also offer a version in which the rabbit *beta-globin* polyA has been replaced with a WPRE and a polyA from bGH (pCAGEN-WPRE-bGH-DEST) (Fig. 5E). Next, we expanded the versatility of the pCAGEN-DEST vector by creating a set of bicistronic pCAGEN-DEST-IRES-myristoylated FP vectors (myr-BFP, myr-EGFP and myr-mKate2) to enable simultaneous fluorescence cell surface labeling. Whereas translation initiates in the first cistron at the 5′ capped mRNA for the protein of interest (supplied by a pME vector), translation of the second cistron (the myr-FP contained in the Destination backbone) occurs in a cap-independent manner mediated by the IRES. These may prove useful in cases in which the amino acids added from a 2A peptide fusion could alter the function of the upstream or downstream protein. Transient transfection of pCAGEN-3×-NLS-mCherry-IRES-myr-EGFP into mammalian cells confirmed the functionality of these

tools (Fig. 5F). Furthermore, co-transfection of pCAGEN-rtTA3 (Tet-On)-IRES-nls-EGFP-WPRE-bGH-pA and a TRE-mCherry reporter confirmed constitutive expression of nls-EGFP, while mCherry was only evident in the presence of doxycycline (Fig. 5G-I).

### Dre/rox Destination vectors for single Entry cloning
We created a Dre recombinase reporter Destination vector in which a CAG promoter drives expression of a nuclear-localized, V5 epitope-tagged RFP variant, mKate2 (nls-mKate2-V5), followed by a rabbit *beta-globin* polyA sequence, all flanked by two *rox* recombination sites, and a downstream Gateway recombination cassette (pCAGGS-*rox*-nls-mKate-V5-stop-*rox*-Dest, or pCAG-rsr-nK-DEST). An FRT flanked PGK-NEO cassette was then added to enable selection of stable clones via neomycin resistance (pCAGGS-rsr-nK-DEST-FRT-PGK-NEO-FRT). Recombination with a pME-myr-TagBFP2-FLAG generated a *rox*-dependent, red-to-blue Dre reporter vector, pCAGGS-*rox*-3×NLS-mKate2-V5-stop-*rox*-myr-TagBFP2-FLAG-FRT-pgk-NEO-FRT (pCAGG-rsr-nKmB) (Fig. 6A). Co-transfection with Dre recombinase revealed efficient switching of nuclear RFP signal for membrane-localized BFP (Fig. 6B). The entire CAG-rsr-nKmB cassette can be removed via AscI to PacI for insertion into murine gene targeting vectors, as shown by our generation of a *Rosa26^rsr-nKmB* reporter mouse (Devine et al., 2014). As an alternative to the *Rosa26* locus, we also provide a DEST vector for the *Hipp11* (*Igs2*) locus on chromosome 11 in the gene desert between *Eif4enif1* and *Drg1* (Hippenmeyer et al., 2010).

To facilitate the creation of novel Dre reporter vectors, we generated a 3×NLS-mCherry-V5-2×STOP insert flanked by *rox* recombination sites (pME-*rox*-nls-mCherry-V5-2×Stop-*rox*). Three-way LR recombination of pME-*rox*-3×NLS-mCherry-stop-*rox* and p3E-Vhh-actin-mNeonGreen into pCS2-Dest2 (Villefranc et al., 2007) generated pCS2-*rox*-nls-mCherry-2×-stop-*rox*-Actin-Vhh-mNeonGreen. Transient transfection confirmed robust mNeonGreen labeling of the actin cytoskeleton only in the presence of Dre recombinase (Fig. 6C,D).

### Tissue-specific Destination vectors for single Entry cloning
We also provide Destination vectors to drive expression in a spatially and/or temporally restricted manner. pβ-MyHC-promoter DEST drives cardiomyocyte-restricted, embryonic expression using the murine β myosin heavy chain (*Myh7*) promoter (Rindt et al., 1993), while pxMLC2-pro-DEST drives cardiomyocyte expression using the *Xenopus laevis myosin light chain 2* (*myl7*) promoter (Breckenridge et al., 2007; Latinkic et al., 2004). pMEF2c-AHF-DEST, which we previously validated (Devine et al., 2014), drives expression within anterior second heart field progenitors of the early mouse embryo via an enhancer of *Mef2c*.

We also created p*GLAST*-DEST for driving expression within the radial glia lineages (cortical neurons, astrocytes, oligodendrocytes and olfactory bulb interneurons) (Chen and LoTurco, 2012; Kim et al., 2003). Collectively, these plasmids should expand the toolkit available to researchers interested in either *in vitro* or *in vivo* neuronal and cardiovascular research.

### Tet-regulated Destination vectors for single Entry cloning
Many doxycycline-inducible (e.g. 'Tet-On') Gateway mammalian plasmids exist, such as the pInducer suite (Meerbrey et al., 2011) and pRAM (Sorensen et al., 2016). However, many Tet-regulatable vector systems show activity in the absence of doxycycline (Meyer-Ficca et al., 2004; Johansen et al., 2002; Mizuguchi and Hayakawa, 2001), likely due to excessive accumulation of the reverse

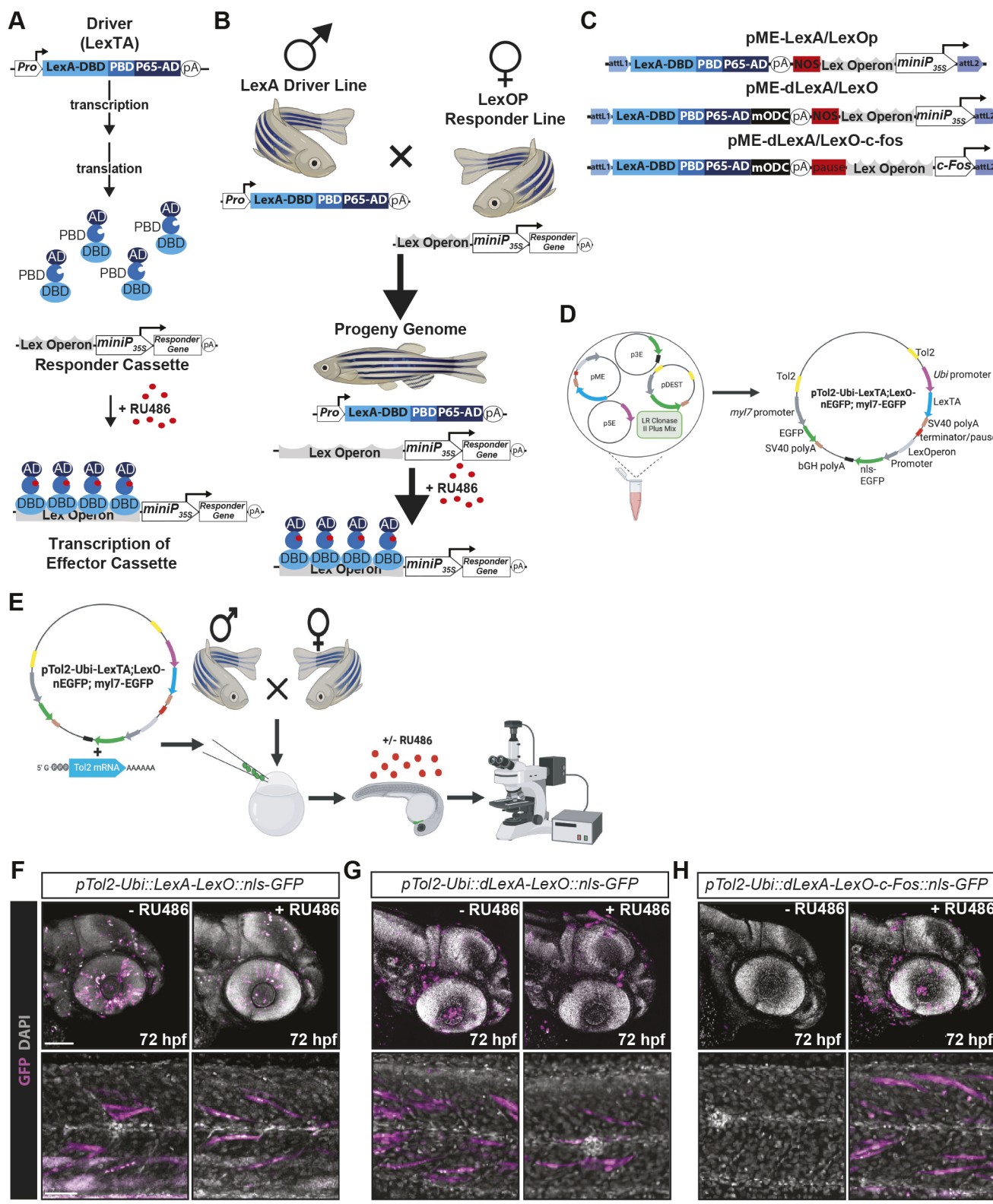

**Fig. 4.** See next page for legend.

tetracycline transactivator and the resultant non-specific activation of the TRE-linked promoter. To develop a tightly controlled Tet-On system for use in mammalian cells, we focused on the Tet-On, *piggyBac* compatible pB-TA-ERN vector (Addgene plasmid #80474) (Kim et al., 2016b) (Fig. 7A).

*piggyBac* transposons have an enormous cargo limit (>200 kb), inserting at TTAA sites with little selectivity, and when combined with pBase transposase (Cary et al., 1989) can drive long-term expression in mammalian cells (Burnight et al., 2012; Li et al., 2011, 2013a,b,c; Nakanishi et al., 2010). As a first step, we

**Fig. 4. The LexA/LexO inducible system.** (A) The mifepristone (RU486)-inducible LexA/LexO gene expression system consists of two components. The first is a 'driver', in this case the LexA transcriptional activator (LexTA). LexTA is a fusion between the Lex DNA-binding domain, the progesterone ligand-binding domain and the transcriptional activator domain of NF-κB p65. This hybrid, or chimeric, transcription factor binds to the synthetic steroid, mifepristone (RU486) and binds the second component, an 'effector cassette', composed of a synthetic lexA operator (LexO) upstream of a cauliflower mosaic virus 35S minimal promoter ($miniP_{35s}$) in a ligand-dependent manner to drive transcription of the downstream gene. (B) Typically, the driver and effector are maintained in different lines and only combined in *trans*. (C) Modification of the LexA system by addition of a destabilization domain from mouse ornithine decarboxylase (mODC) to the LexTA to prevent accumulation of the transcriptional activator and replacement of the cauliflower $miniP_{35s}$ promoter with the minimally active mammalian *c-Fos* promoter and increasing the number of LexOperon binding sites from four to six. (D) LR Clonase II reaction scheme used to generate the various constructs tested. (E) Experimental design for testing of the various constructs by transient transgenesis. (F-H) Confocal images following treatment with either vehicle (DMSO) or RU846 (2 μM) from 24 to 72 hpf. Note the leaky transgene induction of this 'all in one' system unless LexTA is destabilized and the $miniP_{35s}$ is replaced with the *c-Fos* minimal promoter (compare the far-left and middle panels to the far-right panels). Scale bars: 100 μm (top panels, head); 50 μm (bottom panels, somites).

exchanged the potent CMV-IE enhancer and distal full-length rat EF1α promoter in pB-TA-ERN for a transcriptional pause site to prevent local integration effects, and then inserted a full-length human EF1α promoter (hEF1α). Next, we added an amino terminal PEST degradation domain from murine ornithine decarboxylase (Odc1; here termed mODC) (Li et al., 1998) to rtTA to reduce the half-life of the transactivator. Using the resulting plasmid, pB Tet-On 2.0 (pB_TetOn_DEST_hEF1a_mODC-rtTA_IRES_ NEO), as a substrate we engineered a third variant in which the full-length human EF1α promoter was replaced with the short, intron-less human EFS promoter (Schambach et al., 2006) to generate pB-Tet-On 3.0 (pB-TetOn-DEST-EFS-mODC-rtTA-IRES-NEO). We next quantified how these alterations affected Tet-On system kinetics.

Western blotting confirmed robust mCherry induction in pB-Tet-On- and pB-Tet-On 2.0-transfected cells after doxycycline addition, while pB-Tet-On 3.0 showed inferior expression (Fig. 7B). Quantification of luciferase activity confirmed that pB-Tet-On and pB-Tet-On 2.0 had similar fold induction, although pB-Tet-On 2.0 exhibited less variation, while pB-Tet-On 3.0 was less robust. Notably, basal activity of both pB-Tet-On 2.0 and 3.0 was reduced compared to pB-Tet-On (Fig. 7C). Examination of protein lysates before addition and following doxycycline withdrawal confirmed that these newer variants exhibit better on/off kinetics than previous designs (Fig. 7D).

To complement these tools, we engineered a *piggyBac* compatible, Tet-Off vector in which expression of the insert requires Cre-mediated recombination (pB-Tet-Off-FLEX-EFS-mODC-tTA) (Fig. 7E). Transfection of pB-Tet-Off-FLEX-Luc-P2A-H2A-mCherry-EFS-mODC-tTA with and without Cre followed by luciferase analysis confirmed activity was only detected in the presence of Cre and absence of doxycycline (Fig. 7F). Moreover, western blotting verified the tight transcriptional control of this system, as expression was undetectable 16 h post-addition of doxycycline (Fig. 7G).

We extended this combinatorial approach to generate an adeno-associated virus (AAV) Tet-Off vector wherein $TRE_{tight}$ is upstream of a Destination cassette flanked by a pair of double inverse-oriented *loxP* sites, a WPRE and bGH polyA, followed by a transcriptional pause site and minimal EFS promoter driving expression of a destabilized tTA (pAAV-Tet-Off-FLEX-DEST) (Fig. 7H). As expected, using this

vector expression of a mScarlet-P2A-KRAS$^{G12D}$ insert was evident in the presence of Cre, but undetectable following addition of doxycycline (Fig. 7I,J).

### *piggyBac* Destination vectors for MultiSite Gateway cloning

To maximize the flexibility of the MAGIC kit in mammalian systems, we constructed *piggyBac*-compatible MultiSite Gateway Destination plasmids to facilitate integration of the ITR-flanked transgene into the genome (Fig. S5A). MAGIC follows the same recombination logic as zebrafish *Tol2* MultiSite Gateway toolkits to ensure compatibility with these resources. Briefly, three Entry clones (p5E, pME and p3E) integrate in the 5′-to-3′ direction into a single *piggyBac* ITR-flanked Destination vector (Fig. 8A). Two of these Destination vectors contain a downstream SV40 promoter driving expression of a selectable marker to generate stable cell lines (pB-Dest-NEO; pB-Dest-PURO) (Fig. S5B). A third variant (pB-Dest) lacks a stable selection cassette, while a fourth (pB-2×Ins-Dest) contains flanking pairs of the 250-bp core chicken hypersensitive site-4 (cHS4) insulator from β-globin to protect integrated transgenes against heterochromatin propagation and silencing (Chung et al., 1997; Recillas-Targa et al., 2002; Yusufzai and Felsenfeld, 2004; West et al., 2004).

To illustrate the utility of these reagents, we generated a *piggyBac* ITR-flanked cargo plasmid in which *GLAST* promoter-driven expression of oncogenic HRAS$^{G12V}$ within radial glia and astrocytes is dependent upon Cre-mediated excision of an upstream *loxP*-flanked stop cassette (pB-*Glastpro-loxP*-3×Stop-*loxP*-mCherry-T2A-HRAS$^{G12V}$-SV40-polyA) (Fig. 8A). *In utero* electroporation (IUE) of this cargo plasmid along with a *GLAST*-driven hyperactive *piggyBac* transposase (hyPBase) (Yusa et al., 2011), together with a self-excising iCreI plasmid (pCAGEN-se-iCreI-HA), resulted in stable transgene expression in radial glia descendants and frank glioma, as evidenced by extensive upregulation of reactive astrocytes shown by immunohistochemistry for anti-glial fibrillary acidic protein (GFAP) (Fig. 8B-D).

### Cre-compatible, *Tol2* Destination vectors for MultiSite Gateway cloning

By combining a heat shock promoter for temporal control of Cre recombinase expression with a tissue-specific promoter upstream of a floxed stop cassette, we created an 'all-in-one' vector that obviates the need for crossing multiple transgenic lines, as required in the two-component zebrafish HOTcre system (Hesselson et al., 2009). Here, *Tol2* transposon elements, which when used in conjunction with *Tol2* transposase mRNA enable robust and efficient transgenesis in zebrafish (Kawakami et al., 2004), flank the MultiSite Gateway cassette in the 5′-to-3′ direction, followed by an SV40 late polyA signal sequence and a pair of 250-bp chicken β-globin cHS4 insulators to prevent the interaction between the distal enhancer and the upstream promoter (Recillas-Targa et al., 2002; Ryu et al., 2007) (Fig. S3C). In the opposite direction, a heat shock-inducible *hsp70* promoter (Halloran et al., 2000) drives transcription of *Cre.zf1* – a codon optimized variant for expression in zebrafish (Horstick et al., 2015). The 5′ and 3′ halves of the open reading frame of zCre are separated by an artificial intron (*CreI.zf1* or *zCreI*), ensuring that Cre expression (and thus recombinase activity) is restricted to eukaryotic cells (Kaczmarczyk and Green, 2001). zCreI was fused C-terminally to TagBFP to monitor expression, as well as an mODC peptide to prevent Cre accumulation and non-specific recombinase activity. In addition to the parental variant (*pTol2-Dest; hsp70-zCreI-BFP-mODC*), we also provide a variant lacking the mODC peptide (*pTol2-Dest; hsp70-zCreI-BFP*) where zCreI-TagBFP accumulation

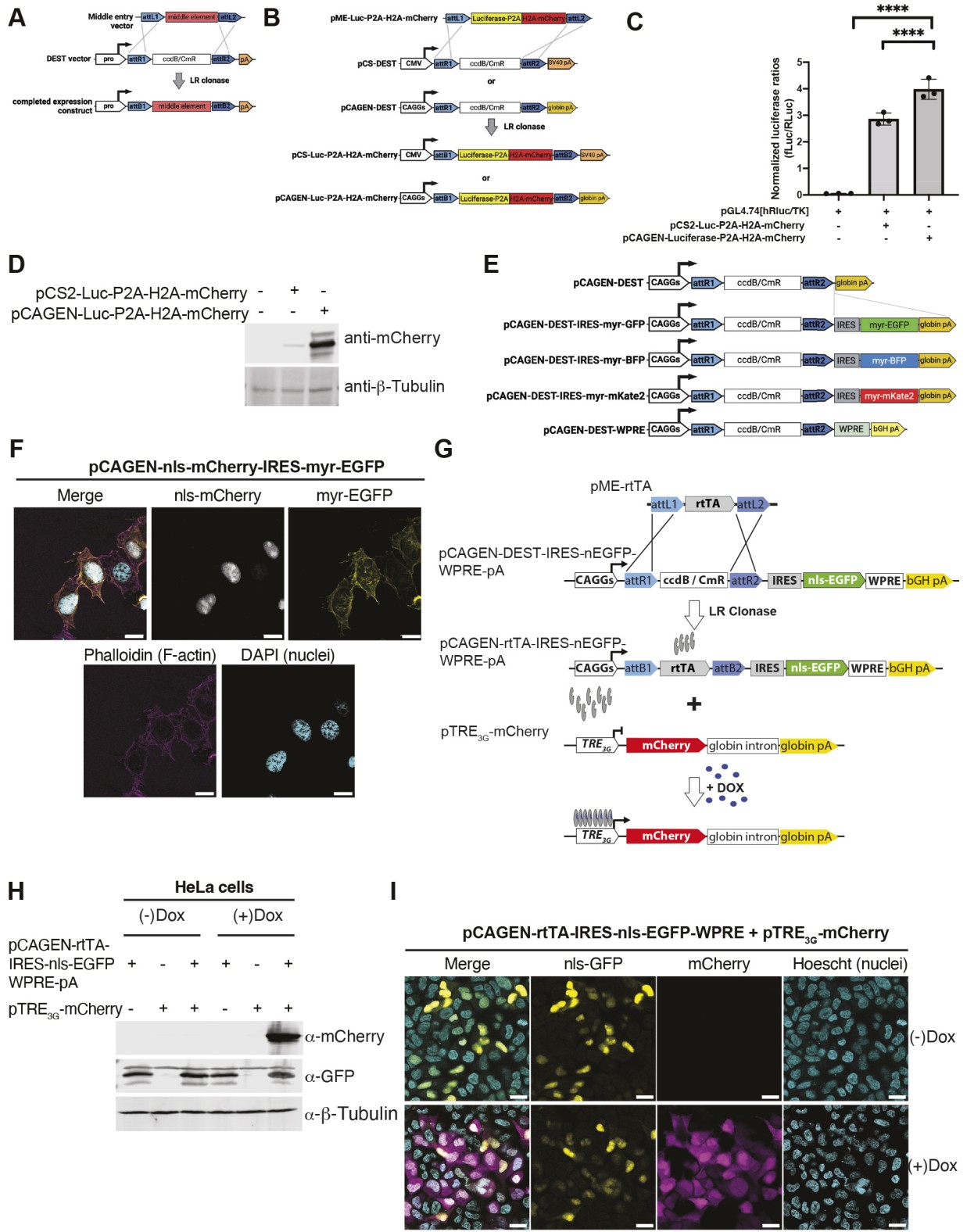

**Fig. 5.** See next page for legend.

may enable more robust recombination. Finally, we created a modified, tissue-specific version of this second-generation HOTcre 'all in one' plasmid where the zebrafish *fli1a* enhancer/promoter (*fli1ep*) drives endothelial-specific expression of zCreI-mTagBFP (*pTol2-Dest2a-2×Ins; fli1ep-zCreI-BFPn*) (Fig. S3C).

As a proof of concept, we generated a ubiquitin promoter-driven green-to-red switch vector where a heat shock promoter controls expression of Cre recombinase (p5E-*Ubi-loxP*-EGFP-*loxP*-nls-mCherry;*hsp70*-zCreI-BFP-mODC) (Fig. 8E). Heat shock at 24 h post-fertilization (hpf) switched expression from a cytosolic EGFP to

**Fig. 5. pCAGEN-based Destination vectors.** (A) Schematic for pME-compatible, Gateway Destination mammalian expression vectors. (B) Comparison of the pCS2-backbone based vector (pCSDest) developed by the Lawson laboratory (see Villefranc et al., 2007), which utilizes a CMV promoter with the CMV immediate early enhancer cassette and the SV40 polyA sequence, with the new pCAGEN-Dest vector that contains the synthetic CAG promoter and β-globin polyA from pCAGGs (see Matsuda and Cepko, 2004). Example of an LR reaction used to generate pCS-Luciferase-P2A-H2A-mCherry and pCAGEN-Luciferase-P2A-H2A-mCherry. (C) Luciferase assays confirm increased transcriptional activity using the pCAGEN-Dest backbone than pCSDest. Error bars represent s.d. ****$P<0.0001$ (one-way ANOVA). (D) Western blots and densitometry for mCherry protein confirm the luciferase analysis. (E) A suite of other dual reporter (IRES-fluorescent reporter backbones) pCAGEN-DEST vectors, as well as a vector with a WPRE element for increased transcript stability. (F) Confocal images of pCAGEN-nls-mCherry-IRES-myr-EGFP show dual fluorescence activity. (G) Schematic for pME-rtTA recombination with pCAGEN-DEST-IRES-nls-EGFP-WPRE-pA and co-transfection with a pTRE-mCherry reporter. (H) Western blot for detection of mCherry, GFP and a loading control (β-tubulin) following transfection in HeLa cells of Tet-On pCAGEN-rtTA-IRES-nls-EGFP-WPRE-pA and a pTRE-mCherry reporter confirm induction of mCherry in the presence but not absence of doxycycline (Dox). (I) Epifluorescence imaging following transfection of the indicated construct in the presence and absence of Dox confirm selective induction of the TRE-mCherry reporter. Scale bars: 20 μm.

nuclear mCherry by 48 h later, establishing the utility of this 'all-in-one' vector for spatially and temporally controlled gene expression (Fig. 8F,G).

### I-Sce Destination vectors for MultiSite Gateway cloning

*I-SceI*, an intron-mediated homing endonuclease (Jacquier and Dujon, 1985; Macreadie et al., 1985), has an 18-bp recognition sequence expected to occur only once in $7\times10^{10}$ random bases and is unlikely to cut genomic DNA. Accordingly, co-injection of the *I-SceI* meganuclease with a construct flanked at the 5′ and 3′ ends by the corresponding recognition sites leads to enhanced stable transgenesis compared to circular, supercoiled plasmid, and it also favors low copy number insertion within a single genomic locus (Thermes et al., 2002). Additionally, the *I-SceI* meganuclease system circumvents the limited persistence of episomal plasmid DNA and overcomes low germline transmission rates due to unequal integration of episomal DNA fragments into the host genome (Etkin and Pearman, 1987; Westerfield et al., 1992; Collas and Alestrom, 1998; Culp et al., 1991; Lin et al., 1994; Stuart et al., 1988). We supply I-SceI-Dest-pA, where I-Sce sites flank a MultiSite Gateway cassette, followed by a polyA signal to allow for assembly of p5E, pME and p3E inserts within the same backbone.

### A relational database for plasmid management

To accompany this plasmid resource, we developed a fully modifiable, relational database using FileMaker software for maintaining electronic records of all plasmids and their salient features, including PubMed identifiers, laboratory source, Addgene information, promoters, open reading frames, various uses of the plasmid, and other notes (Figs S7-S10). We provide a step-by-step guide to using and modifying the database (Movies 1, 2), as well as a freely available empty database (supplementary information).

### DISCUSSION

MultiSite Gateway cloning, combined with Tol2 transposon-mediated transgenesis, revolutionized the zebrafish field (Fowler et al., 2016; Kwan et al., 2007; Villefranc et al., 2007). The MAGIC toolkit, which is compatible with existing zebrafish Gateway kits, should facilitate transgenesis in other eukaryotic systems,

particularly those that utilize *piggyBac*- or I-Sce-mediated transgenesis. The utility of this toolset spans cultured mammalian cells, to murine and chicken embryos, to adult mice via AAV and lentiviral transduction, and, of course, teleost species.

Herein, we provide tools for robust, constitutive expression in cultured mammalian cells with our pCAGEN suite of Destination plasmids. We also supply dual reporter IRES-driven systems for monitoring transfection efficiency, and a suite of fluorescent probes for visualizing actin dynamics, organelles and specific subcellular compartments, as well as cell cycle status. This collection of ubiquitous and tissue-specific promoters, Destination vectors, and novel middle Entry fluorescent reporters, will facilitate transgenesis *in vitro* and *in vivo*.

Simultaneously, we also generated Gateway-compatible tools for optogenetically (light) and chemogenetically regulated (doxycycline, RU486) gene expression and validated their use not only in cultured mammalian cells, but also in mice and zebrafish. We extended these chemically inducible gene expression systems by making them Cre dependent, and expanded the utility of these tools by making *piggyBac*-, I-Sce-, lentiviral- and AAV-compatible Destination vectors. At the same time, we generated novel Cre-dependent tools, as well as reagents for the robust SSR Dre, for *in vitro* and *in vivo* studies. These advances, as well as our validation of novel Tol2-dependent 'all-in-one' Cre/*loxP* and LexA/LexO vectors for use in zebrafish, open new possibilities for manipulating gene expression in both time and space in the developing and adult zebrafish, mouse and chick, as well as in numerous *in vitro* mammalian cell-based systems.

### Study limitations

Given the breadth of reagents supplied in this toolkit, some components were not validated during the preparation of this article. However, many novel components were, and previously validated tools were assumed to retain their functionality after being transferred to the Gateway system [i.e. the microRNA containing EGFP intron plasmids, a platform which we previously validated (Fish et al., 2008), or the Dre/*rox* system, which we validated in mice (Devine et al., 2014)].

A FileMaker subscription may be prohibitive for some laboratories (as of spring 2025 it costs $17.50 per month, per user, for a locally hosted annual license; $624 for a perpetual license). However, this software provides an excellent graphical user interface, stable environment and operational support with continued updates and can host multiple different databases (for an example, see Cantu Gutierrez et al., 2019). Importantly, the database architecture itself is open access, as is every field and form, indeed every relationship within this relational database is modifiable by the end user. Similarly, the cost of plasmids at Addgene (currently $85 per plasmid) may be viewed as prohibitive, but a robust archive of these tools is essential for preventing their loss and contamination. Overall, these limitations pale in comparison to the value of providing this extensive suite of Gateway-ready tools.

### MATERIALS AND METHODS
### A relational database for plasmid management

Due to the sheer volume of plasmids generated for this toolkit, and with the mandate for central lab organization and more robust data management from the National Institutes of Health, we developed a relational database using FileMaker software for maintaining electronic records of all plasmids and their salient features, including PubMed identifiers, Addgene information, promoters, open reading frames, various uses of the plasmid, and other notes (Fig. S7). Crucially, electronic files (such as those generated by APE or SnapGene) can be posted directly into database records, as can illustrations for

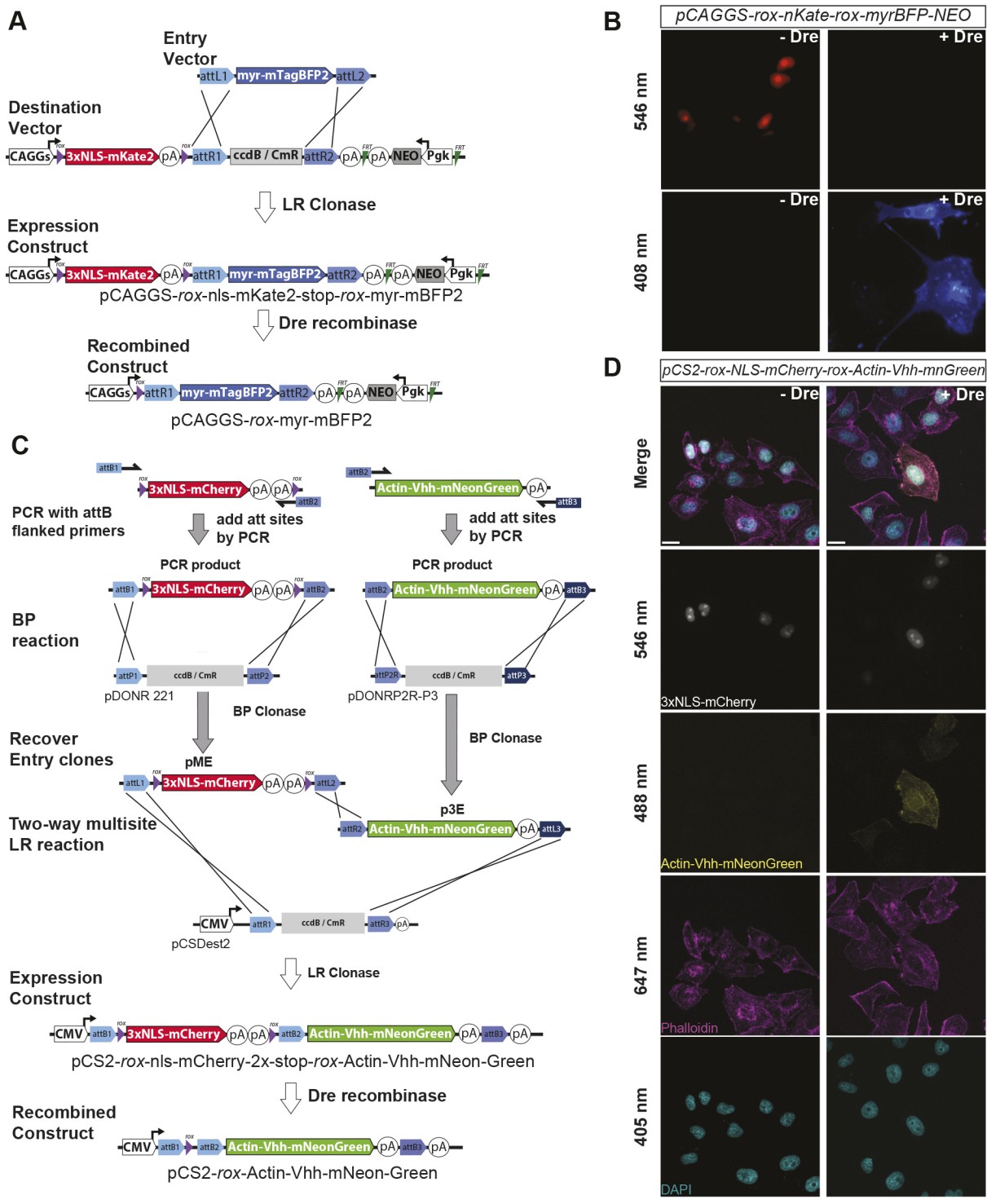

**Fig. 6. The Dre/*rox* recombinase system.** (A) Schematic for a pME-compatible Gateway Destination vector where a CAG promoter drives expression of a *rox*-flanked nuclear mKate2 reporter, and a pME cassette, such as one containing myristoylated mTagBFP2 (myr-mTagBFP2), can be inserted downstream. Upon Dre-mediated recombination of the *rox*-flanked expression cassette, mKate2 and the transcriptional stop sequence (polyA) are removed, and the downstream myr-mTagBFP2 cassette is expressed. (B) Validation of the pCAG-nuclear-Kate-myr-BFP (nKmB) reporter following co-transfection in HEK-293T cells with and without Dre recombinase. (C) Generation of a novel dual Dre/*rox* reporter, where 3×NLS mCherry and a 2× stop cassette are flanked by *rox* recombination sites in a pME vector, and a novel p3E construct containing an actin nanobody (Vhh) fused to mNeonGreen are inserted distally into a pCSDDest2 vector. (D) Validation of the switch reporter activity in the presence and absence of Dre recombinase. Scale bars: 20 μm.

maps of the plasmid (e.g. as .jpeg or .png files). All fields and container fields within the database are modifiable, as the package is presented as an open-source tool that can be easily modified by anyone familiar with FileMaker software. This database can be shared by an entire laboratory either on one centralized computer, or through multiple computers provided they each have access to FileMaker Pro software or that the database is hosted on a local

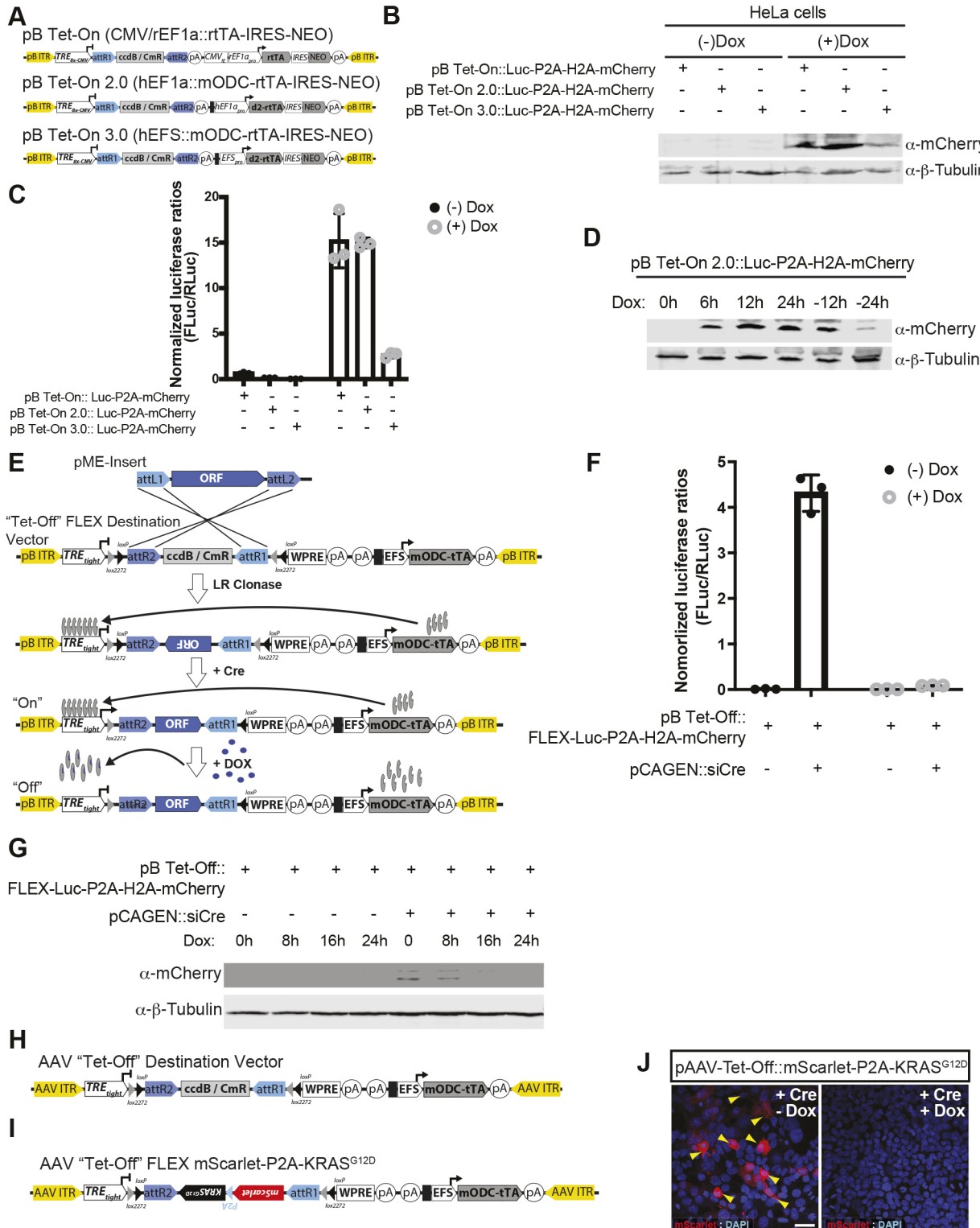

**Fig. 7. A suite of Tet-On and Tet-Off Gateway vectors for eukaryotic studies.** (A) Three variants of Tet-On vectors, with modifications to the promoters driving rtTA expression (CMV_IE and rat EF1a versus human EF1a alone versus an EF1a short or EFS minimal promoter), along with regular rtTA versus a destabilized rtTA (d2rtTA) to prevent accumulation of the transcriptional activator. (B) Western blot for mCherry and a loading control, β-tubulin, confirm varying levels of doxycycline (Dox) induction for each of the three variants following transient transfection in HeLa cells. (C) Luciferase reporter activity following transient transfection in HeLa cells. Error bars represent s.d. (D) Kinetics of Luciferase-P2A-H2A-mCherry expression shown by western blotting following addition and withdrawal of Dox. (E) A binary, Cre-dependent, Tet-Off Destination vector design. (F) Validation and kinetics in HeLa cells. Error bars represent s.d. (G) Luciferase reporter assay validating Tet-off, Cre-dependent construct in the presence/absence of Dox and Cre. (H,I) A similar AAV-based FLEX or DIO approach for Cre-dependent, Tet-Off expression (H) and an example vector, AAV-Tet-Off-FLEX-mScarlet-P2A-KRASG12D (I). (J) Validation of Cre-dependent expression and Dox-induced silencing of transcription following transient transfection in HEK-293T cells. Arrowheads indicate KRAS-expressing, mScarlet-positive, Cre-recombined cells. Scale bar: 20 μm.

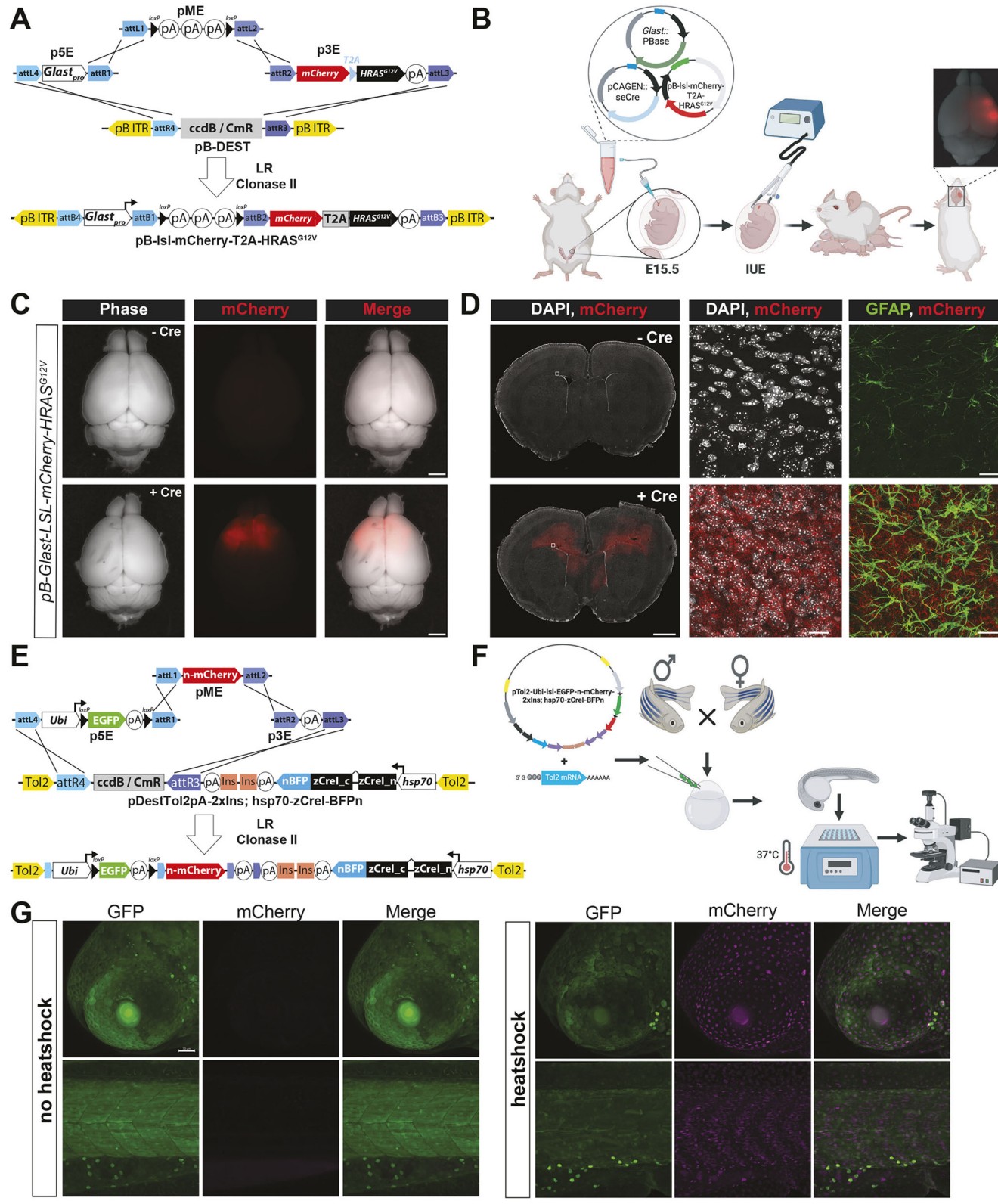

**Fig. 8. Novel Cre/lox Gateway-compatible vectors for vertebrate transgenesis.** (A) Diagram of the three-way LR cloning reaction and the resulting *pB-Glast-3×STOP-mCherry-T2A-HRAS^G12V*. (B) Schematic for *in utero* electroporation (IUE) in mice followed by visualization of mCherry-positive, HRAS-driven glioblastoma in mice. (C) Wholemount phase and epifluorescence images from representative brains. (D) Coronal sections from these same brains showing clear mCherry-positive tumor cells in the +Cre animal. White boxed area is magnified in panels to the right, showing confocal images from sections stained for mCherry (tumor), GFAP (reactive astrocytes) and DAPI (nuclei). (E) Three-way LR recombination reaction used to create *pTol2-2×Ins-Ubi-lox-EGFP-stop-lox-NLS-mCherry; Hsp70-zCreI-BFP-NLS*. (F) Experimental workflow for heat shock. (G) Example from a stable *Tg(Ubi-lsl-EGFP-NLS-mCherry;Hsp70-zCreI-BFP)* line following either room temperature incubation, or heat shock, for 1 h and then imaged at 72 hpf. Scale bars: 2 mm (C); 20 μm (D); 50 μm (G).

computer or server (we host the database using a dedicated Mac Mini with its own IP address). Importantly, many of the data entry fields in this database correspond to those directly used by Addgene. Dropdown lists for container fields can be modified by any user, or entirely new lists or fields created, to provide a fixed vocabulary for categories such as plasmid use (for instance, zebrafish transgenesis, subcloning, mammalian expression, Gateway subcloning, etc.), and promoter type (e.g. CAGGS, EF1a, CMV, etc.). Other standard categories, such as whether there are bacterial or DNA stocks available, can be created and feature yes/no dropdown tabs (Fig. S8).

Data from these various entry fields can be exported as a tab delimited file to Excel to facilitate sharing or depositing plasmids at Addgene or other plasmid repositories, or to other laboratories, and individual PDFs of any record can also be generated directly in FileMaker and exported for sharing information about the plasmid of interest. The various entry fields we have created, such as plasmid use or promoter, can be modified by individual users (or removed entirely), as can the layout of the form, the size of data entry fields/containers, and indeed the entire underlying architecture of the system is open to user manipulation (Figs S9, S10). A typical data entry workflow is included for reference (Movie 1), as is a video showing navigation and search functionality (Movie 2). Of note, the more detailed the description within the entry fields and containers, the better the search functionality will operate for finding plasmid records. An empty database, with one complete plasmid entry included for a reference, is provided (supplementary information) and is also available on our laboratory website at https://www.wythelab.com/wythe-lab-databases.

### Creating Entry clones

Entry vectors can be generated by multiple methods. We typically use PCR to add the desired *attB* sites to the insert to be cloned, or synthesize gene fragments directly with flanking *attB* sites, and then perform a BP reaction to catalyze recombination with the *attB* flanked insert and *attP* sites within a Donor plasmid, which ultimately generates an *attL*-flanked, insert-containing Entry plasmid (Fig. S1). If PCR fails, or gene synthesis is not practical, Entry vectors are available from Addgene for conventional restriction enzyme-based cloning (e.g. p5E-MCS, Addgene plasmid #26029), and we supply a few novel variants of these vectors in the MAGIC kit. Additionally, DNA ligase-free cloning can be used with directional topoisomerase-based (TOPO) cloning kits to insert PCR products into an *attL*-containing Entry vector.

### BP cloning of Entry vectors

Entry vectors were created following the recommendations described in the MultiSite Gateway manual (Invitrogen), with minor modifications. In most cases, *att* sites were added onto the ends of DNA fragments by PCR using either Q5 High-Fidelity DNA Polymerase (NEB, M0491) or Platinum Taq DNA Polymerase High Fidelity (Life Technologies, 11304011).

For p5Es (using pDONR P4-P1R), the forward PCR primer contained an *attB3* site preceding the template specific sequence, while the reverse primer contained a reverse *attB1* site following the template-specific sequence. For generation of pMEs (using pDONR221 or pENTR1a), the forward primer contained an *attB1* site, while the reverse primer contained an *attB2* site. For generation of p3Es (using pDONR P2R-P3), the forward primer contained an *attB2* site, while the reverse primer contained a reverse *attB3* site. Table S1 lists all *att* sequences used in these PCR reactions.

Following PCR, the amplified products were either PCR purified (QIAGEN, 28104), or gel extracted using the QIAquick gel extraction kit (QIAGEN, 28704). As storage of purified PCR products, even at −20°C, decreases recombination efficiency (Kwan et al., 2007), 150 ng of PCR product was immediately combined with 150 ng of pDONR, then mixed with 2 μl of BP Clonase II enzyme mix (Invitrogen, 11789020) and TE (pH 8.0) in a total volume of 10 μl and incubated overnight at room temperature. The following day, the reaction was terminated by the addition of 1 μl of Proteinase K (2 μg/μl) and incubated at 37°C for 10 min, then transformed into chemically competent, recombination-deficient Stbl3 bacteria (Invitrogen, C737303) and plated on LB agar plates containing the appropriate antibiotic (in most cases kanamycin, unless otherwise noted). Recombinant clones were screened by colony PCR using primers that flanked the insert, and all positive clones were then confirmed by Sanger

sequencing across the entire Entry cassette and through the flanking *att* arms. Note that the manufacturer suggests combining the PCR insert and pDONR vector at a 1:1 molar ratio, using roughly 50 fmol of each component for a BP reaction.

### Vector generation and LR cloning

For details regarding generation of all p5E, pME, p3E and Destination vectors, see supplementary Materials and Methods. For a comparison of the amino acid sequences of mScarlet variants, see Fig. S11. For a comparison of the amino acid sequences of the StayGold variants, see Fig. S12. Alternatively, an automated molecular calculator for determining the molar ratios is available online (Mosimann, 2022).

### Mammalian cell culture

HeLa (ATCC, CCL-2) and HEK-293T (ATCC CRL-3216) cells were purchased from ATCC and used for all mammalian cell culture validation experiments. Cells were grown in 10% fetal bovine serum, DMEM (Thermo Fisher Scientific, 11966025), 1× Penicillin-Streptomycin (Thermo Fisher Scientific, 15140148), non-essential amino acids (Thermo Fisher Scientific, 11140035) and GlutaMAX (Thermo Fisher Scientific, 35050061). Cells were routinely passaged using TrypLE Express reagent (Thermo Fisher Scientific, 12605010) prior to reaching 90% confluency. Cell lines were routinely tested for contamination, particularly *Mycoplasma*, prior to use in experiments.

### Luciferase analysis

HeLa cells ($3 \times 10^4$ cells/well) were seeded in a 96-well white polystyrene plate (Thermo Scientific, 15492) coated with fibronectin from bovine plasma (1:400 dilution of a 1 mg/ml stock) (Sigma-Aldrich, F1141) the day before transfection. The following day, the cells were co-transfected with 100 ng of a firefly reporter vector and 20 ng of a normalizing *Renilla* reporter (pRL-TK) using PolyJet transfection reagent (SignaGen Laboratories, SL100688) according to the manufacturer's instructions. For tet-inducible firefly reporters, cells were treated with doxycycline (1 μg/ml; Sigma-Aldrich, D3072), or vehicle control, for 24 h prior to measuring luciferase activity using the Dual-Glo Luciferase Assay System (Promega, E2920) on a BioTek Cytation-1 Cell Imaging MultiMode Reader (Agilent) 48 h after transfection. Each assay was performed with technical replicates, over three experimental days (biological triplicates). Reporter activity was calculated (using the average of each technical replicate) to determine the ratio of firefly to *Renilla* activity across all three experiments. The data were then plotted using GraphPad software (Prism).

All plasmids have been deposited at Addgene to enable rapid, easy distribution to the biological community. For updates, please see our lab webpage: https://www.addgene.org/Joshua_Wythe/.

### Mammalian cell culture confocal imaging

For imaging, the day before transfection HeLa ($1 \times 10^4$ cells/well) (ATCC, CCL-2) or HEK-293T (ATCC CRL-3216) cells were seeded in 4-well Nunc Lab-Tek II chamber slide (Thermo Scientific, 154453) coated with poly-L-lysine (Sigma-Aldrich, P6282) (0.1 mg/ml). The following day, the cells were transfected with various fluorescent reporter plasmids (0.36 μg DNA/well) using the PolyJet transfection reagent (SignaGen Laboratories, SL100688) according to the manufacturer's instructions. Forty-eight hours later, the cells were fixed with 4% paraformaldehyde (PFA)/1× PBS at room temperature for 10 min, then permeabilized with 0.2% Triton X-100/1× PBS for 10 min and then blocked in 3% bovine serum albumin/1× PBS for 1 h at room temperature. Then, the cells were co-stained with DAPI (Thermo Fisher Scientific, D1306) (a 5 mg/ml stock was diluted to a working concentration of 300 nM), and in some cases Alexa Fluor 647 nm Phalloidin (Thermo Fisher Scientific, A22287) (1:200) or Rhodamine (TRITC, 565 nm) Phalloidin (Thermo Fisher Scientific, R415) (1:400) in 3% bovine serum albumin/1× PBS for 1 h at room temperature, and then mounted with Prolong Diamond Mounting Media (Thermo Fisher Scientific, P36965). For nuclear staining, cells were either incubated with Hoechst 33342 (1 μg/ml) (Thermo Fisher Scientific, H1399) for 30 min in a 37°C, 5% $CO_2$ incubator or post-fix stained for 20 min with DAPI (1 μg/ml) (D1306, Thermo Fisher Scientific). For HaloTag labeling, cells were

incubated with 200 nM of JFX646-HaloTag Ligand (a generous gift of Janelia Fluor Dyes) for 15 min in 37°C incubator prior to fixation with 4% PFA/1× PBS. Images were captured using a Leica SP8 confocal microscope with the laser power set between 1 and 5%, using a 63× oil lens (NA=1.4). Captured z-stack images (1024×1024 pixels) were processed using LAS software and then exported to ImageJ (Schneider et al., 2012).

## Western blotting
HeLa cells (1×10⁶ cells/well) were seeded in a 6-well plate the day before transfection. The following day, cells were transfected with 1 µg of plasmid DNA using PolyJet transfection reagent (SignaGen Laboratories, SL100688). Twenty-four hours after transfection, cells were treated with 1 µg/ml of doxycycline (or vehicle control) for 24 h (unless otherwise noted in the figure legend). Then, cell lysates were extracted 48 h post-transfection in 300 µl of radioimmunoprecipitation assay (RIPA) buffer (25 mM Tris-HCl pH 7.6, 150 mM NaCl, 1% NP-40, 1% sodium deoxycholate, 0.1% SDS) plus Halt™ Protease and Phosphatase Inhibitor Cocktail (Thermo Fisher Scientific, 78430), and diluted to 1× using 4× Laemmli buffer (Bio-Rad, 1610747) with 2-mercaptoethanol. Samples were then boiled for 5-10 min at 100°C and 20 µl of each sample was loaded onto a 12% SDS-PAGE gel for electrophoresis and then transferred to Immobilon-FL PVDF membrane (Millipore, IPFL00010) using a Power-Blotter Semi-dry transfer system (Thermo Fisher Scientific). Membranes were then incubated in blocking buffer (50% Intercept PBS blocking buffer (LI-COR, 927-70001) diluted in 1× PBS/0.1% Tween 20 (PBST) for 1 h at room temperature, then incubated in blocking buffer with rabbit anti-mCherry (Invitrogen, PA5-34974; RRID:AB_2552323; 1:1000) or anti-GFP (Rockland Immunochemicals, 600-401-215; RRID:AB_828167; 1:1000) and mouse monoclonal anti-β-tubulin (Invitrogen, 32-2600; RRID: AB_86547; 1:3000) overnight at 4°C. The next day, the blots were washed in PBST and then incubated with secondary antibodies diluted in blocking buffer for 1 h at room temperature. The secondary antibodies used were goat anti-rabbit IgG (H+L) Secondary Antibody DyLight™ 800 4× PEG (Invitrogen, SA5-35571; RRID:AB_2556775; 1:10,000) and goat anti-mouse IgG (H+L) Cross-Adsorbed Secondary Antibody DyLight™ 680 (Invitrogen, 35519; RRID:AB_1965956; 1:10,000). Antibody binding was detected using the Odyssey Imaging System (LI-COR Biosciences).

## Zebrafish experiments
Zebrafish protocols were approved by the Animal Care Committee at Baylor College of Medicine and the University of Virginia. Embryos were collected from timed matings and raised in 1× E3 (5 mM NaCl, 0.17 mM KCl, 0.33 mM CaCl₂, 0.33 mM MgSO₄) at 28.5°C and staged according to time post-fertilization and morphology (Kimmel et al., 1995). Beginning at approximately 8 hpf, embryos were incubated in 1× E3 with 0.003% (w:v) (200 µm) 1-phenyl 2-thiourea (PTU) to prevent pigment formation. The following transgenic lines were utilized: Tg(kdrl:GFP)^{s843} (Jin et al., 2005) and Tg(kdrl:mCherry)^{ci5} (Proulx et al., 2010). The following line was created: Tg(Ubi-lsl-EGFP-nls-mCherry; hsp70-zCreI-BFP)^{va100}.

## Tol2-mediated transgenesis
Tol2-mediated transgenesis (Kawakami et al., 2004) was performed as previously published (Wythe et al., 2013). Briefly, NotI-linearized, gel-purified pCS2FA-transposase (Kwan et al., 2007) plasmid was used as a template for in vitro transcription using the mMessage mMachine SP6 kit (Ambion, AM1340) to generate capped, Transposase mRNA. The amplified mRNA was purified and subsequently ethanol precipitated, resuspended in TE diluted 1:10 with ddH₂O, then quantified. A mixture of 100 ng of purified DNA (phenol:chloroform purified and resuspended in resuspended in TE diluted 1:10 with ddH₂O), 125 ng of Tol2 Transposase mRNA, 1 µl 0.8% Phenol Red/0.1 M KCl/pH 7.0 and ddH₂O in 10 µl of total volume was loaded into a glass microcapillary and 1 nl injected directly into the cell of one-cell-stage zebrafish embryos.

## RU486/LexA-LexO induction and imaging
pTol2-based LexA/LexO nls-EGFP reporter plasmids were co-injected along with transposase mRNA into one-cell-stage AB zebrafish embryos as described above. At 24 hpf, embryos with cardiac-specific expression of the cmlc2::GFP reporter cassette were collected and evenly distributed into two

wells of a 6-well plates for each tested construct. A 10 mM stock of mifepristone/RU486 (Biotechne/Tocris, 1479) in pure DMSO was thawed from −20°C and then diluted 1:5000 (or an equivalent amount of DMSO for the vehicle control treatment) to make a 2 µM working solution in 1× E3 with PTU (to inhibit pigmentation). Embryos treated with RU486, or vehicle control, were then incubated in the dark at 28.5°C. At ~72 hpf, embryos were rinsed in ice-cold 1× PBS, then fixed in 4% PFA/1× PBS in the dark overnight at 16°C with gentle agitation. The following day, embryos were washed three times in 1× PBS at room temperature, then permeabilized by three 20-min washes of 0.5% Triton X-100/1× PBS at room temperature on an orbital shaker, before being incubated in 300 nM DAPI (Thermo Fisher Scientific, D1306)/1× PBS at room temperature for 40 min in the dark with gentle agitation. Embryos were then washed three times in 1× PBS, 5 min per wash, briefly post-fixed in 4% PFA/1× PBS, rinsed in 1× PBS, and then mounted in 1% low-melt agarose in 1× PBS on a 35 mm glass-bottom tissue culture dish (Mattek, P35G-1.5-14-C) and covered in 1× PBS prior to being imaged on a Leica SP8 confocal microscope using an HC PL APO 10×/0.4 objective. pTol2-Ubi-LexA-LexO-CAMV35S-nls-EGP-pA-injected embryos were imaged with the 405 nm channel (DAPI) at 8.54% power and a digital gain of 800. The 488 nm channel (GFP) was imaged at 10.8% power and a digital gain of 800. pTol2-Ubi-dLexA-LexO-CMV35S-nls-EGFP-pA- and pTol2-Ubi-dLexA-LexO-c-fos-nls-EGFP-pA-injected embryos were imaged with the 405 nm channel at 2.95% power and a digital gain of 692. The 488 nm channel was imaged at 2.4% power and a digital gain of 800. All images were captured with a pinhole size of 1 AU (53 µm). All z-stacks were collected using a 5 µm step size and exported image sequences were processed using ImageJ and figures made using Adobe Photoshop and Illustrator.

## Heat shock
To test the pDestTol2-Ubi-lsl-EGFP-nls-mCherry; hsp70-zCreI-BFP construct (Fig. 8), fresh Tg(Ubi-lsl-EGFP-nls-mCherry; hsp70-zCreI-BFP)^{va100} embryos were collected from timed spawnings and then incubated overnight at 28.5°C in 1× E3/0.003% PTU. At 24 hpf, embryos were transferred to 1× E3/0.003% PTU pre-warmed to 37°C and maintained at that temperature for 30 min. Subsequently, embryos were then rinsed once in 28.5°C 1× E3 with PTU, and then incubated at that temperature until the following day (48 hpf) when another 30-min heat shock at 37°C was performed. Embryos were returned to 28.5°C until 72 hpf and then fixed overnight in 4% PFA/1× PBS at 4°C with gentle agitation and protection from light. Fixed embryos were embedded in 1.5-2.0% low-melt agarose dissolved in either 1× PBS or 1× E3 and embedded sagittally on a Mat-Tek coverglass 35 mm Petri dish.

Embryos were imaged on a Zeiss LSM 880 using a Plan-Apochromat 20×/ 0.8 M27 objective. The red channel (561 nm) was imaged at 60% laser power, a digital gain setting of 632, and with a pinhole of 57 µm for both heat shock and non-heat shock conditions. The green channel (488 nm) was imaged using 12% laser power, with a gain of 607 and a pinhole of 57 µm. Widefield images were captured on a Zeiss V16.Zoom macroscope.

## TAEL photoinduction and imaging
Post-injection, embryos were raised in the dark, with blue light blocked by covering the incubator door, and transmitted light on the microscope for sorting of embryos or aging, with red gel filters (Pangda-Gel Filter Colored Overlays, purchased through Amazon). For photoinduction of TAEL activity, embryos were anesthetized and immobilized in 1× Tricaine/1× E3/0.003% PTU/0.5% low-melt agarose and then illuminated on a Nikon ECLIPSE Ti2 equipped with a Yokogawa W1 spinning disk unit with a Plan apo lambda 20× lens, using 488 laser at 50% power, and 1 s exposures at 1 min intervals for approximately 10 min with the aperture reduced to target illumination to the region of interest. Images were acquired using a PFS4 camera.

## Murine experiments
All mouse experiments were approved by the Institutional Animal Care and Use Committee at Baylor College of Medicine and the University of Virginia. For all experiments, noon on the day a vaginal plug was discovered was considered embryonic day (E) 0.5, the day of birth was considered postnatal day (P) 0, and all adult mice were at least 8 weeks of age. Mice were housed with access to food (normal chow diet) and water ad libitum on

a 12-h light/12-h dark cycle at 21°C and 50-60% humidity. Timed matings were set up at the end of the workday. Appropriately timed pregnancies were used as described below.

## IUE

IUE was performed as previously described (Carlson et al., 2021). Briefly, the uterine horns were surgically exposed in a pregnant dam (CD-1 IGS strain) at E15.5 and the embryos were injected with a DNA cocktail containing the following plasmids: (1) pB-*Glast-loxP*-3×STOP-*loxP*-p3E-mCherry-T2A-HRAS$^{G12V}$, (2) a *piggyBac* (pB) helper plasmid with the glial- and astrocyte-specific promoter, *Glast*, driving expression of PB transposase (pGlast-PBase), and (3) a self-inactivating, improved Cre with an intron (pCAGEN-se-iCreI-HA, JDW 1226).

The PBase helper plasmid promotes stable integration of the cargo fluorescent reporter vector, which indelibly labels all descendant cells, allowing one to visualize tumors over time by fluorescence. Following injection of the glioma-inducing cocktail (2.0 µg/µl pGLAST-PBase; 1.0 µg/µl all other plasmids) into the lateral ventricle of each embryo, embryos were electroporated six times at 100-ms intervals using BTW Tweezertrodes connected to a pulse generator (BTX 8300) set at 33 V and 55 ms per pulse. Voltage was applied across the entire brain to allow uptake of the constructs. The uterine horns were placed back in the cavity, and these dams developed normally, but their electroporated offspring featured malignancies postnatally, as the tumor suppressor-deficient cells expanded. IUE tumors were harvested at P21 from both male and female mice and processed as described below for immunohistochemistry.

## Immunohistochemistry

Brains from both male and female electroporated mice were harvested and fixed in 4% PFA/1× PBS overnight at 4°C in the dark with gentle agitation on an orbital shaker. The following day, brains were washed in 1× PBS and mounted in 1% agarose. Coronal sections (35 µm thick) were collected using a compresstome (World Precisionary Instruments, VF-300-Z) and subsequently mounted on Superfrost Plus microscope slides (Fisher Scientific, 22-037-246). Sections were dried at 50°C for 2 h, washed once with 1× PBS, and then dried at 60°C for 1 h. A barrier was then drawn around each section using an ImmEdge Hydrophobic Barrier Pen (Advanced Cell Diagnostics, 310018). Samples were rehydrated with ddH$_2$O for 5 min before incubating them for 1 h in blocking buffer (1× PBS/10% donkey serum/0.5% Triton X-100). Mouse monoclonal anti-GFAP 488 (clone GA5) (Thermo Fisher Scientific, 53-9892-82; RRID: AB_10598515; 1:50) and rabbit polyclonal anti-mCherry (Invitrogen, PA5-34974; RRID:AB_2552323; 1:100) were then diluted in blocking buffer and sections were incubated with primary antibody in blocking buffer solution overnight at room temperature. The following morning, samples were washed three times in blocking buffer. Sections were then incubated for 2 h at room temperature in goat anti-rabbit Alexa Fluor 568 (Invitrogen, A-11011; RRID:AB_143157) diluted in blocking buffer (1:200). Sections were then washed five times in blocking buffer, and then mounted with ProLong Gold Antifade Mountant with DAPI (Invitrogen, P36931).

Wholemount images post-dissection were taken with an AxioZoom V16 using a 2.5× objective with 20% overlap for tiled reconstruction of the brain. Tiled images were stitched together using Zen (Blue edition) software. Sections were imaged using a Leica SP8 confocal microscope with laser power set between 1 and 5% (GFP, mCherry) and ~20% (DAPI), using a 63× lens (NA=1.4). *z*-stack images (1024×1024 pixels) were processed using LAS software and then exported to ImageJ.

## LiCre induction

Mice were anesthetized by injection of a ketamine/dormitore mixture and were maintained under anesthesia using vaporized isoflurane with O$_2$. Mice were then affixed to a stereotaxis apparatus synced to Angle Two software for coordinate guidance. *Ai14* Cre reporter mice (Madisen et al., 2010) (12 weeks old) were bilaterally injected into the anterior mouse motor cortex (M1) and a unilateral injection into the posterior motor cortex (M1, AP=+0.7, DV=−1.8, ML=±2; posterior M1, from bregma: AP=−1.0, DV=−1.3, ML=±1.1) with 690 nl per hemisphere of a pLenti-CMV-LiCre. Concurrently, mice were bilaterally implanted with 230 µm silica fiber optic

implants (200 µm core with NA=0.22, RWD *R*-FOC-L200C-22NA) and situated 0.1 mm above the viral injection site. Fiber optic implants were held in place by a cap made from adhesive cement (C&B Metabond Quick! Cement System, Parkell) for initial base, and crosslinked flash acrylic (Yates-Motloid, 44115 and 44119) for headcap. Mice were allowed to recover, and viral transduction to occur, for 2 weeks before photoactivation. For stimulation, each animal was tethered to a dual fiber optic cord (Doric Lenses) attached to a 473 nm laser source (CrystaLaser, CL-2005) and then mice were chronically stimulated with trains of blue light (30 mW, 10 ms pulses, 5 or 20 Hz, 5 s trains, 30 s intervals, for 1 h total). As a control group, the contralateral side, which was also injected and implanted, was not stimulated. Mice were euthanized 1 week later, and brains collected for free-floating cryosectioning and counter-staining with DAPI to label all nuclei. In some cases, mice we perfused with fluorescent lectin prior to euthanasia to label the cerebrovascular endothelium. Sections were then mounted and imaged using a confocal microscope (Leica SP8).

Twenty-four hours before transfection, HEK-293T cells were seeded into 12-well plates (Thermo Fisher Scientific, 07-200-82) at a density of 3×10$^5$ cells/well. The following day, cells were co-transfected with 500 ng of a Cre-mediated switch reporter (pCMV-*lox*-zsGreen-stop-*lox*-mCherry, JDW 514) and either self-inactivating Cre (pCAGEN-si-CreI, JDW 1226) or light-activated Cre (pCAGEN-LiCre, JDW 1220). Transfection complexes were made using PEI (1:3 ratio of DNA:PEI) (Polysciences Inc., 23966). PEI:DNA complexes were allowed to form for 15 min and then 50 µl of OptiMEM (Thermo Fisher Scientific, 31985062) was gently added to the mixture and then the PEI/DNA/Opti-MEM mixture was added, dropwise, to the cells. The transfection mixture was left on cells for 6 h and then the media was aspirated and replaced with DMEM/10% serum/1× non-essential amino acids/1× Penicillin-Streptomycin. At 24 h post-transfection, some wells were illuminated with blue light (460 nm, 1.1 watts) for a duration of 30 min on/30 min off, for a total of 24 h. Another group of cells was maintained in the dark. At 48 h post-transfection, both groups were fixed in 2% PFA/1× PBS for 15 min and mounted with ProLong Gold with DAPI (Thermo Fisher Scientific, P36935). Images were captured using a Leica SP8 confocal with laser power set between 5 and 10%, using a 20× lens (NA=0.4). *z*-stack images (512×512 pixels) were processed using LAS software and then exported to ImageJ.

## Acknowledgements
We thank Dr Margot L. K. Williams (Baylor College of Medicine) for her assistance with the TAEL experiments, and Dr Benjamin Deneen (Baylor College of Medicine) for the gift of *piggyBac* ITR flanked cargo plasmid templates and the pGlast promoter template.

## Competing interests
The authors declare no competing or financial interests.

## Author contributions
Conceptualization: W.P.D., J.D.W.; Data curation: W.B.G., W.D.T., W.P.D., Y.Z., G.L., J.O.-G., J.D.W.; Formal analysis: W.B.G., Y.Z., J.O.-G., J.D.W.; Funding acquisition: W.D.T., G.L., M.S., B.R.A., J.D.W.; Investigation: W.B.G., Y.Z., O.E.R., J.C., J.O.-G., M.S., W.D.T., G.L., L.E.M., W.P.D.; Methodology: O.E.R., J.C., J.O.-G., B.R.A., W.P.D.; Resources: B.R.A., J.O.-G., M.S., W.D.T., L.E.M., E.F., C.-B.C., K.M.K., W.P.D.; Validation: W.B.G., Y.Z., J.O.-G., M.S., W.D.T., G.L., W.P.D., J.D.W.; Visualization: W.B.G., Y.Z., O.E.R., G.L., J.D.W.; Writing – original draft: J.D.W.; Writing – review & editing: W.B.G., Y.Z., J.O.-G., B.R.A., W.P.D., J.D.W.

## Funding
This work was supported by grants from the National Institutes of Health (5T32GM088129-10 to W.D.T.; T32HL007284 to G.L.; 1R01HL159159 to W.P.D. and J.D.W.) and the Cancer Prevention and Research Institute of Texas (RP200402 to J.D.W.), a Harrison Undergraduate Research Award at the University of Virginia (to M.S.), and funding from the School of Medicine, University of Virginia (J.D.W.). The funders played no role in the study design, data collection and analysis, decision to publish, or preparation of the manuscript. Open Access funding provided by University of Virginia. Deposited in PMC for immediate release.

## Data and resource availability
The FileMaker Plasmid Database can be downloaded at https://www.wythelab.com/wythe-lab-databases and all plasmids may be obtained from Addgene (https://www.

addgene.org/Joshua_Wythe/). All other relevant data and details of resources can be found within the article and its supplementary information.

## Peer review history

The peer review history is available online at https://journals.biologists.com/dev/lookup/doi/10.1242/dev.204308.reviewer-comments.pdf

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
