## [Peer Review File · Development (Cambridge, England)]

MultiSite Assembly of Gateway Induced Clones (MAGIC): a flexible cloning toolbox for use in vertebrate model systems.

William B. Gillespie III, Yuwen Zhang, Oscar E. Ruiz, Juan Cerda III, Joshua Ortiz-Guzman, Michelle Sherman, Williamson D. Turner, Gabrielle Largoza, Lili E. Mosser, Esther Fujimoto, Chi-Bin Chien, Kristen M. Kwan, Benjamin R. Arenkiel, W. Patrick Devine and Joshua D. Wythe
DOI: 10.1242/dev.204308

Editor: Steve Wilson

Review timeline

Original submission:	13 August 2024
Editorial decision:	21 October 2024
First revision received:	9 July 2025
Editorial decision:	9 September 2025
Second revision received:	18 September 2025
Accepted:	1 October 2025

Original submission

First decision letter

MS ID#: dev.204308

MS TITLE: Multisite Assembly of Gateway Induced Clones (MAGIC): a flexible cloning toolbox with diverse applications in vertebrate model systems.

AUTHORS: William Gillespie; Yuwen Zhang; Oscar Ruiz; Juan Cerda III; Joshua Ortiz-Guzman; Michelle Sherman; Williamson Turner; Gabrielle Largoza; Lili Mosser; Esther Fujimoto; Chi-Bin Chien; Kristen Kwan; Benjamin Arenkiel; Walter Patrick Devine; Joshua Wythe

Dear Joshua,

I have now received all the referees' reports on the above manuscript, and have reached a decision. The referees' comments are appended below, or you can access them online: please go to:

As you will see, the referees recognise the value of the tools/resources you present but they have significant concerns about the way that the study is presented. Among the issues raised, primarily by reviewer 2, is that there is not appropriate or sufficient acknowledgment of prior work on related methods and reagents - it is absolutely essential that all such issues are fully addressed in a revised manuscript. For instance, you should always indicate where you have developed a new tool/reagent versus tweaked/modified an existing one. The reviewers also consider that the study should be presented much more concisely focussing on maximising the ability of the readership to understand and access reagents and reducing other aspects of the text including the justification for building the toolbox of reagents. Consequently the study requires a significant re-write - our limit for papers is usually 7000 words and your resubmission should be close to this (and definitely under 7700 words). One further essential change is to use an openly accessible database and not Filemaker for presenting data - the filmmaker software would not be available to many potential users.

If you are able to revise the manuscript along the lines suggested above and by the reviewers, I will be happy receive a revised version of the manuscript. Please also note that Development will normally permit only one round of major revision. If it would be helpful, you are welcome to contact us to discuss your revision in greater detail. Please send us a point-by-point response indicating your plans for addressing the referees' comments, and we will look over this and provide further guidance.

Please attend to all of the reviewers' comments and ensure that you clearly highlight all changes made in the revised manuscript. Please avoid using 'Tracked changes' in Word files as these are lost in PDF conversion. I should be grateful if you would also provide a point-by-point response detailing how you have dealt with the points raised by the reviewers in the 'Response to Reviewers' box. If you do not agree with any of their criticisms or suggestions please explain clearly why this is so.

Reviewer 1

SUMMARY OF THE ADVANCE MADE IN THIS PAPER AND ITS POTENTIAL SIGNIFICANCE TO THE FIELD

The publication "Multisite Assembly of Gateway Induced Clones (MAGIC): a flexible cloning toolbox with diverse applications in vertebrate model systems" by Gillespie et al. describes the generation of a wealth of new plasmids using the Gateway cloning platform. The authors validate some of the new plasmids in different model organisms or cultured cells. The newly generated plasmids will be for sure valuable for the scientific community and a great resource.

SUGGESTIONS TO AUTHORS

The main question, I feel, is how much of the paper is redundant and could be shortened. For instance, Figure 1 is very similar to Figures in Villefranc et al. and to figures in the Gateway manual. I am not sure how necessary it is to explain how to make entry clones and recombination reactions, as this can be found in the Gateway manual as well. It might also be beneficial to summarize the available constructs in a figure with all their components drawn out as the authors, for example, do for the pCAGEN-based Destination vectors in Figure 6 instead of explaining the constructs in the text.

Reviewer 2

SUMMARY OF THE ADVANCE MADE IN THIS PAPER AND ITS POTENTIAL SIGNIFICANCE TO THE FIELD

Gillespie and colleagues present a massive collection of plasmid vectors for transgenesis, geared mainly towards mammalian models with zebrafish-tested components added to the mix. The work builds in two decades of vector collections available to the field and expands these in contemporary ways that are likely of wide-spread interest. The authors provide several proof-of-concept experiments and figures that help gauge applications and utility of individual plasmids and combinations.

Overall, while certainly of widespread interest to the field, and documented with at times excellent imaging, the manuscript is a challenging read and describes several reagents that need to be scrutinized more closely.

Write-ups of vector or reagent collections are inherently challenging, yet especially Development has a long-standing track record of publishing accessible and widely acknowledged resource papers, which should provide ample reference to tackle a revised version.

The current write-up is excessive and will benefit from re-visiting and re-emphasizing a more pragmatic approach akin to other published vector collections. The technical aspects of individual applications are well-presented and executed in general, despite the inherent challenge of gauging robustness of the presented applications with limited examples (a general problem of tool papers).

More challenging is the apparent re-branding, re-labeling, not to say downright re-purposing, of already existing reagents labeled as new in this manuscript. These instances should be revisited throughout as they blur the publication record and prior efforts in the field.

SUGGESTIONS TO AUTHORS

Major Points

1) Concerning the text, the current manuscript is unwieldy at best and and excessively wordy (if not rambling) in individual sections. The text reads more like a book chapter (or manifesto, rather) than a manuscript that aims to be community-accessible. Streamlining the messaging should be doable and will shorten the text, i.e. reducing sections expressing sentiments ("...we also hope that...") instead of scientific messaging.

* Already in the introduction, the authors outline dramatic cloning challenges that might be how they perceive molecular cloning, but that the reviewer argues are hardly general sentiments (i.e. deeming restriction enzyme cloning "be incredibly time consuming and laborious" - anyone doing Multisite or Gibson or Golden Gate will call that their methods, too). The authors then present multisite gateway as a way out of a perceived misery. However, multisite cloning is hardly new and has been embraced by the *C. elegans* and later the zebrafish community (including by co-authors on the work!) to a great extent. The lengthy write-up to present multisite gateway as life-saver is therefore hardly warranted or appropriate. Overall, the introduction reads like a detached take on a general technique that should be drastically revisited in the reviewer's view. The authors can't take credit for Multisite Gateway as such, so why push this?

* The authors provide excessive detail on how multisite cloning works, but omit other, at times critical details (i.e. not just any LR Clonase will do, has to be LR Clonase II Plus for optimal multisite reactions; vector ratios for cloning that drastically improve efficiency are being glanced over; etc.). Similarly, in their results the authors introduce BP reactions to clone entry vectors - TOPO-based cloning kits including pENTR5' and pENTR/D have been mainstays of the community since well over a decade and greatly facilitate entry vector generation. These are neither discussed nor considered.

* Individual sections are highly imbalanced in their length and level of detail. For instance, tissue-specific expression vectors are only minimally explained, and mammalian and zebrafish elements are mixed without too much explanation. While certainly challenging to balance all information content, the authors are encouraged to revisit the individual write-ups to streamline details and length.

2) The linked website for all vector maps was offline at time of review, making it impossible to look at vector maps and to compare to existing vectors (see also below).

3) A challenging aspect of the current write-up and presentation is the inclusion of several reagents that seem at best like minimally altered versions of pre-existing vectors, or at worst re-branded plasmids previously generated by other labs in the field. Obvious examples include:

a) "p5E-Ubi (JDW 804), which contains a ~3.5 kb insert spanning the promoter, exon1, and intron 1 of the zebrafish ubiquitin locus for constitutive expression in *Danio rerio*" - this is re-cloning of the zebrafish ubiquitin promoter from a more complex vector into a gateway 5'vector, which seems identical to the Zon lab's widely used ubb 5' vector pCM206 available on AddGene (Mosimann et al., 2011). Also, why not include mouse or human ubiquitin promoters instead, given that these have long-standing history in the field?

b) The SV40 poly A vector (JDW461) sounds like #302 from the original Tol2 kit (Kwan et al., 2007), which is puzzling given the first-author of the original study and late PI are on the manuscript as co-authors.

c) Vectors JDW 755, 760, and 761 sound again like vectors already introduced in Kwan et al., 2007, specifically #394 and #395 that are the most commonly used Tol2 backbones in the zebrafish field. While laudable that the authors generated several reagents that were not accessible for gateway in the past, the reviewer wonders how the labs who generated individual reagents will take the inclusion of these (i.e. TetO, iCre, TAEL components, shorter or slightly altered promoters from other labs, etc.). Individual components used for vectors are also not always referenced and

explained. This makes the entire write-up present the work as a grab-bag of parts the authors liberally cobbled together.

Minor points:

- 1) Individual transgene graphics seem heterogeneous in their presentation and layout (e.g. figure 10's schematics are different than previous figures) and could be homogenized to increase accessibility
- 2) Overall, figures could be condensed to reach a more accessible 7-8 figure format.
- 3) Species of intent or depicted in the tests should be more clearly labeled and detailed.

Reviewer 3

SUMMARY OF THE ADVANCE MADE IN THIS PAPER AND ITS POTENTIAL SIGNIFICANCE TO THE FIELD

The manuscript details the development of a large number of gateway compatible clones for use in vertebrates to complement and enhance the existing Tol2 kit. While there is not really a conceptual advance here the range and breadth of new components are considerable and a valuable community resource for researchers across disciplines and model systems.

My principle comment is that while a number of the vectors have been tested, the vast majority have not been tested. Clearly with such a large repertoire of potential combinations of promoters etc testing everything is not possible, however this means the functionality of a number of the plasmids and combinations remains untested. e.g. the addition of the mODC peptide to alter the stability of zCre in the zebrafish "hotCRE" variants and the miRNA vectors

The toolkit will be available through addgene making it very accessible to the community, however the filmmaker resources are limited to those who have a annual subscription to filmmaker which comes at a not significant cost and may limit access to this aspect of the tool.

SUGGESTIONS TO AUTHORS

Specific comments and questions below

The manuscript is long and detailed. Some of the summarisation of the existing gateway/tol2kit could be condensed since this is published and widely used for many years

The text from p34 moves over into p36 (figure 7 in between), the end of 34 is the end of a sentence but p36 this starts in the middle of a sentence, so I can't tell what is being said. p40 destabilised tTA already abbreviated

typos - combing should be combining
NK-kB/p65 should be NF-kB/p65 I think

Figures

Figure 1

I'm not sure this is needed. individual figures throughout the manuscript have this detail as well and the gateway system is well described both by the company and more specific to this work in the tol2kit wiki page and manuscript. I suggest moving to supplemental

Figure 2

The lifeact fluorophores seem to label fewer cells than phalloidin. Is this expected for this system?

Figure 3

While this figure demonstrates the utility of the optogenetically controlled Cre, as far as I can tell it does not utilise new components developed in the MAGIC toolkit. Is this the case?

Figure 4

The legend states that 405nm laser is used whereas the methods state 488nm laser. Please clarify

Figure 5

In 5G the promoter with the TRE is labelled TREtight and TRE3G in the figure. I think these the same? Only TRE3G is listed in the plasmid table

Figure 10

pA is labelled in some but not all constructs

In C the mODC is labelled as dest (I think) . Not sure why this abbreviation?

in 10D the construct does not look to have the mODC but in the text it does. Please clarify

Table

pME Vhh mCherry is listed as being shown in figure 2. I can't see this in figure 2

Some icre constructs are labelled as being codon optimised - which species are the optimised for?

Some of the tissue specific dest vectors do not say which species their promoter is derived from

It might be helpful given the last 2 columns to have a "species" column added to the table

First revision

Author response to reviewers' comments

REVIEWER 1:

The publication "Multisite Assembly of Gateway Induced Clones (MAGIC): a flexible cloning toolbox with diverse applications in vertebrate model systems" by Gillespie et al. describes the generation of a wealth of new plasmids using the Gateway cloning platform. The authors validate some of the new plasmids in different model organisms or cultured cells. The newly generated plasmids will be for sure valuable for the scientific community and a great resource.

We thank the reviewer for their positive feedback and acknowledgement of the work required to generate this novel resource.

SUGGESTIONS TO AUTHORS

(1) The main question, I feel, is how much of the paper is redundant and could be shortened. For instance, Figure 1 is very similar to Figures in Villefranc et al. and to figures in the Gateway manual. I am not sure how necessary it is to explain how to make entry clones and recombination reactions, as this can be found in the Gateway manual as well.

We appreciate the Reviewer's point. We have attempted to streamline the main text,

refer to other papers that pioneered this method of cloning, and have moved the Gateway overview figure (old Figure 1) to the supplemental figures. However, as a standalone manuscript, we do explain this extensively in the methods section so that readers have all of the salient details within this single, standalone paper. We hope this satisfies the Reviewer's concerns.

(2) It might also be beneficial to summarize the available constructs in a figure with all their components drawn out as the authors, for example, do for the pCAGEN-based Destination vectors in Figure 6 instead of explaining the constructs in the text.

We are not clear if the Reviewer is referring to every construct made in the manuscript (which would not be possible to contain in a figure given the sheer number of clones and would be unwieldy at best), or if they simply mean the Destination plasmids that we have created (or that are featured in the main figures).

We have added new columns in the Tables to summarize the intended use(s) of the each plasmid, their species of origin, and other salient details, and have also added supplemental figures showing variations of Destination vectors that are offered, while adding additional experimental schematics in the main figures.

If the Reviewer has additional suggestions for schematics to include, we are happy to do so to increase the accessibility of the toolkit.

REVIEWER 2

Gillespie and colleagues present a massive collection of plasmid vectors for transgenesis, geared mainly towards mammalian models with zebrafish-tested components added to the mix. The work builds in two decades of vector collections available to the field and expands these in contemporary ways that are likely of widespread interest. The authors provide several proof-of-concept experiments and figures that help gauge applications and utility of individual plasmids and combinations.

Overall, while certainly of widespread interest to the field, and documented with at times excellent imaging, the manuscript is a challenging read and describes several reagents that need to be scrutinized more closely.

Write-ups of vector or reagent collections are inherently challenging, yet especially Development has a long-standing track record of publishing accessible and widely acknowledged resource papers, which should provide ample reference to tackle a revised version.

(1) The current write-up is excessive and will benefit from re-visiting and re-emphasizing a more pragmatic approach akin to other published vector collections. The technical aspects of individual applications are well-presented and executed in general, despite the inherent challenge of gauging robustness of the presented applications with limited examples (a general problem of tool papers).

Understood. We appreciate the feedback and have attempted to streamline the text.

(2) More challenging is the apparent re-branding, re-labeling, not to say downright re-purposing, of already existing reagents labeled as new in this manuscript. These instances should be revisited throughout as they blur the publication record and prior efforts in the field.

We did not intend to obscure the origin of previously developed tools (or act as if we created an established tool or technology), and all manuscripts were cited in the main text and/or methods section.

In previous zebrafish-oriented Gateway toolkit papers, Kwan et al being a prominent example, shuttling DNA elements that were previously identified and validated in other contexts, such as mammalian cell culture (e.g. SV40 polyA, EGFP, mCherry, etc.), into Gateway compatible backbones was not considered rebranding or repurposing, so we are a bit confused by the Reviewer's point.

In the case of existing zebrafish DNA elements, such as the *cmcl2/myh7* promoter, subsequent papers that cloned it into p5E and pDEST vectors were doing the same thing as we have done herein (i.e. using *fli1* or *hsp70*, etc.). More recently (in this journal), Kemmler and colleagues used a previously validated pineal gland promoter (Asaoka Y. et al., PNAS, 2002; PMID 12438694) to drive fluorescent reporters in a Tol2 backbone (Kemmler CL et al., *Development*, 2023; PMID: 36975217). Given these prominent examples, we hope we our work is being held to the same standard.

We have done extensive work to expand the repertoire of available promoters, middle entry, and 3' clones, as well as destination vectors, while also purposefully expanding beyond Tol2 backbones to provide novel piggybac, AAV, and lentiviral destination plasmids for use in mammalian systems (as well as adding some novel Tol2 Destination plasmids). The substrate for these vectors are each appropriately cited, using their original labels and terminology, and are not "re-branded".

We also optimize several existing plasmids by swapping in new minimal promoters (e.g. exchanging the *b-actin* minimal promoter for *c-Fos* to reduce basal transcriptional readthrough, of CMV/EF1a for EFS, etc.), adding or removing introns (e.g. in the case of Cre), adding degradation tags (mODC elements), and providing several fluorescent reporters that were not included in previous papers (as both pME and p3E clones).

We have also extended many of these tools by making several new all in one versions (hot-cre, LexA/LexO, TAEI, Tet-On/Off, etc.) which advances the field.

We do not see how our approach is distinct from previous papers (i.e. Kwan KM et al., 2007; Fowler DK et al., 2016; Villefranc JA et al., 2007; etc.). Indeed, this approach has been repeated in several Gateway compatible toolkit papers published in this journal (*Development*), and other similar tier journals. At a minimum, the sheer number of plasmids (>100) and scope of validation approaches - mouse electroporation, zebrafish transgenesis, extensive mammalian cell culture - (with the exception of Fowler et al which spanned both mammalian cell culture and zebrafish models) should be adequate for a novel tools and resources paper, provided we have adequately cited and attributed the original tools we adapted (which we assure the Reviewer was our intent).

SUGGESTIONS TO AUTHORS

Major Points

1) Concerning the text, the current manuscript is unwieldy at best and and excessively wordy (if not rambling) in individual sections. The text reads more like a book chapter (or manifesto, rather) than a manuscript that aims to be community- accessible. Streamlining the messaging should be doable and will shorten the text, i.e. reducing sections expressing sentiments ("...we also hope that...") instead of scientific messaging.

We have streamlined the text to make the revised manuscript more accessible to readers and less of a "rambling manifesto".

* Already in the introduction, the authors outline dramatic cloning challenges that might be how they perceive molecular cloning, but that the reviewer argues are hardly general sentiments (i.e. deeming restriction enzyme cloning "be incredibly time consuming and laborious" - anyone doing Multisite or Gibson or Golden Gate will call that their methods, too). The authors then present multisite gateway as a way out of a perceived misery. However, multisite cloning is hardly new and has been embraced by the *C. elegans* and later the zebrafish community (including by co-authors on the work!) to a great extent. The lengthy write-up to present multisite gateway as life-saver is therefore hardly warranted or appropriate. Overall, the introduction reads like a detached take on a general technique that should be drastically revisited in the reviewer's view. The authors can't take credit for Multisite Gateway as such, so why push this?

While the "misery" of restriction-based cloning is not obvious to the Reviewer (and we would note it is *their* opinion that this is on equal footing with Gateway cloning for ease and simplicity), we disagree. After all, if restriction-based cloning is so simple and versatile, then why was Gateway cloning adapted (or Golden Gate cloning, or cold fusion cloning, etc.)? We are trying to encourage the mammalian community to adopt this methodology and thus we conveyed the same advantages that led to its widespread adoption by the zebrafish community (rather than referring readers to another manuscript). Further, we reasoned that many readers outside of the zebrafish field are not familiar with Gateway cloning and its advantages. Thus, we provide a rationale for its use. A passing referring to other manuscripts is, in our opinion, not an adequate introduction.

* The authors provide excessive detail on how multisite cloning works, but omit other, at times critical details (i.e. not just any LR Clonase will do, has to be LR Clonase II Plus for optimal multisite reactions; vector ratios for cloning that drastically improve efficiency are being glanced over; etc.). Similarly, in their results the authors introduce BP reactions to clone entry vectors - TOPO-based cloning kits including pENTR5' and pENTR/D have been mainstays of the community since well over a decade and greatly facilitate entry vector generation. These are neither discussed nor considered.

The materials and methods text for the "LR Cloning" section in the first draft stated that LR Clonase II Plus enzyme mix is used, and it detailed the exact amounts of all plasmids, times for incubation, and other salient details. We have amended the main text to explicitly state this requirement for multisite reactions. In our experience 150 ng of each plasmid, rather than equimolar calculations, is sufficient to recover a suitable number of correct clones. However, to address the Reviewer's concerns we now point readers to existing resources to perform these calculations should they desire to do so.

In the original manuscript we discussed restriction-based cloning, as well as BP cloning, as we supplied novel MCS-based clones for pENTR backbones. In the revised paper we now mention TOPO-based cloning methods for generating p5E, pME, and p3E vectors.

* Individual sections are highly imbalanced in their length and level of detail. For instance, tissue-specific expression vectors are only minimally explained, and mammalian and zebrafish elements are mixed without too much explanation. While certainly challenging to balance all information content, the authors are encouraged to revisit the individual write-ups to streamline details and length.

Understood.

2) The linked website for all vector maps was offline at time of review, making it impossible to look at vector maps and to compare to existing vectors (see also below).

The Addgene upload took much longer than expected, but 144 are available as of July 1, 2025, and the remainder are undergoing quality control at Addgene:

<https://w.addgene.org/browse/article/28248562/>

3) A challenging aspect of the current write-up and presentation is the inclusion of several reagents that seem at best like minimally altered versions of pre-existing vectors, or at worst re-branded

plasmids previously generated by other labs in the field. Obvious examples include:

a) "p5E-Ubi (JDW 804), which contains a ~3.5 kb insert spanning the promoter, exon1, and intron 1 of the zebrafish ubiquitin locus for constitutive expression in *Danio rerio*" - this is re-cloning of the zebrafish ubiquitin promoter from a more complex vector into a gateway 5' vector, which seems identical to the Zon lab's widely used ubb 5' vector pCM206 available on AddGene (Mosimann et al., 2011).

In this particular instance we failed to realize a p5E-Ubi vector was already available/derived from the same parental vector that we had used to generate our p5E Ubi plasmid (the substrate being, as detailed and cited in the materials and methods section of the original draft, *p5E-Ubi-loxP-EGFP-stop-loxP*, Addgene # 27322, from Mosimann et al., 2011). We then validated our re-made version in the hot-Cre system in Figure 10 of the original manuscript.

As far as we can tell, while the Addgene plasmid and ours differ by 2 nucleotides, this is likely because our recent long range nanopore sequencing picked up something different than when Dr. Mosimann's p5E-Ubi / Addgene # 27320 was originally deposited and sequenced at Addgene. As the two vectors overlap exactly at the 3' edge of the BamHI site, we believe we repeated exactly what they had done with the same parental vector (thus inadvertently re-making the vector, as despite the 2 bp discrepancy we believe they are the same). We have removed this plasmid from our manuscript and from Addgene. We sincerely thank the Reviewer for pointing out this oversight out to us.

Also, why not include mouse or human ubiquitin promoters instead, given that these have long-standing history in the field?

In the original manuscript we included three ubiquitous human promoters: p5E-*EF1a*, p5E-*EFS*, and p5E-CAGGs. In the revised submission have added a p5E-human *UBIQUITIN* promoter for use in mammalian studies.

b)The SV40 poly A vector (JDW461) sounds like #302 from the original Tol2 kit (Kwan et al., 2007), which is puzzling given the first-author of the original study and late PI are on the manuscript as co-authors.

As detailed in **Table 4** in the "Origin/Publication" column (far right side of the table), the plasmid is p3E-polyA from Kwan et al., 2007. We have our own plasmid unique ID system in the lab (as shown in the supplemental database we included) to track all cloning/plasmid generation steps. We hoped that listing the plasmid source in the table and citing the relevant manuscript would make clear what we had generated in the paper vs what was derived before (hence the "Wythe lab/this paper", "Kwan lab/this paper" vs "Chien Lab / Kwan et al., Dev Dyn" annotations in the Tables, and also why we don't describe it as being generated in the text of the paper or the materials and methods section).

In the revised manuscript, we have eliminated the internal JDW numbering for all published plasmids (while retaining the far-right side column citing the source of the plasmid). We hope that this, along with the clear annotations in the tables, make the origin of all referenced plasmids clear to readers.

c) Vectors JDW 755, 760, and 761 sound again like vectors already introduced in Kwan et al., 2007, specifically #394 and #395 that are the most commonly used Tol2 backbones in the zebrafish field.

Please see above.

While laudable that the authors generated several reagents that were not accessible for gateway in the past, the reviewer wonders how the labs who generated individual reagents will take the inclusion of these (i.e. TetO, iCre, TAEL components, shorter or slightly altered promoters from other labs, etc.). Individual components used for vectors are also not always referenced and explained. This makes the entire write-up present the work as a grab-bag of parts the authors liberally cobbled together.

As stated above, the goal was to create an extensive toolkit of promoters, site-specific recombinases, fluorescent reporters, and plasmid backbones for adopting this methodology in model systems other than zebrafish (mouse, chick, cell culture models, etc.). While the Reviewer may consider it a “grab-bag”, we consider them common cell and developmental biological tools that will facilitate adoption of this toolkit (which is also directly compatible with existing zebrafish toolkits).

Previous publications detailing the relevant scientific discovery of each tool are clearly cited in our manuscript. We provide references for all plasmids used as templates for generating new vectors, either in the main text or the methods section. Additionally, we cite papers where sequences were obtained from for plasmids that were synthesized (such as for the actin nanobodies, for example), and we directly reference sources of plasmid templates for subcloning and PCR (i.e. Addgene #s, laboratories, etc.). If we have omitted this information in the manuscript (which we assure the Reviewer is an inadvertent, unintentional error), ***please point it out directly and we will make sure the proper citations are in place, as that is not our intent.***

Minor points:

1) Individual transgene graphics seem heterogeneous in their presentation and layout (e.g. figure 10's schematics are different than previous figures) and could be homogenized to increase accessibility

We have changed them to be more consistently formatted in the main figures.

2) Overall, figures could be condensed to reach a more accessible 7-8 figure format.

The manuscript now includes 8 figures in the main text.

3) Species of intent or depicted in the tests should be more clearly labeled and detailed.

The revised Tables list the plasmid origin (species), as do the file descriptions at Addgene for all uploaded plasmids. The figure legend text clearly states the mammalian cell type, while zebrafish figures show the Tol2 system and zebrafish embryos have zoomed out views that alert the reader to the fact the magnified views are from zebrafish. We hope this satisfies the Reviewer's concerns.

REVIEWER 3:

The manuscript details the development of a large number of gateway compatible clones for use in vertebrates to complement and enhance the existing Tol2 kit. While there is not really a conceptual advance here the range and breadth of new components are considerable and a valuable community resource for researchers across disciplines and model systems.

My principle comment is that while a number of the vectors have been tested, the vast majority have not been tested. Clearly with such a large repertoire of potential combinations of promoters etc testing everything is not possible, however this means the functionality of a number of the plasmids and combinations remains untested.
e.g. the addition of the mODC peptide to alter the stability of zCre in the zebrafish "hotCRE" variants and the miRNA vectors.

We apologize for not making it clear, but we previously published the (non-Gateway) *miR-126 EGFP* tools (Fish et al., *Dev Cell*, 2008), as well as the *Dre/rox* system and *Hipp11* clones (Devine et al., *Elife*, 2013) that we adapted for Gateway use in this manuscript.

Within the paper we extensively characterize the various Tet-On systems in pCAGEN in this manuscript, with and without the mODC variants/destabilization sequence (Figure 5), and validate the Tol2 “all in one” new HotCre system (Figure 8), as well as the lox-stop-lox piggybac system (Figure 8), the Dre tools (Figure 6) in vitro (and in vivo, within Devine et al., *Elife*, 2014), the LexA/LexO tools with and without the mODC (Figure 4), the FUCCI plasmids (Figure 2, Fig. S4), and many of the subcellular fluorescent reporter tools (Figure 1, Fig. S3), and the impact of various promoter and mODC modifications on the Tet-On system (Figure 7) by luciferase assay and

western blot.

Thus, many of these tools have been validated using a wide breadth and depth of experimental validation, in multiple models (mice, cell culture, zebrafish), with multiple readouts (imaging, westerns, luciferase assays, recombination assays).

The toolkit will be available through Addgene making it very accessible to the community, however the filmmaker resources are limited to those who have a annual subscription to filmmaker which comes at a not significant cost and may limit access to this aspect of the tool.

FileMaker is \$21/month for 2GB of data storage and 5-10 users to host 3 applications (such as the Plasmid Database). This fee compares to the cost of software like SNAPGENE, MATLAB, PRISM graphpad, the Adobe suite, Microsoft word/powerpoint/excel, and it does not represent a significant financial hurdle (to say nothing of the cost of LR clonase II, BP clonase II, etc.).

The benefits of FileMaker, as we have previously shown for a zebrafish husbandry database we published (Gutierrez AC et al., *PLoS Biol*, 2019) is that it is entirely open source in the sense that the underlying architecture is customizable by users (if a user decides an entry field is worthless, they can easily remove it, if they want additional description fields/categories, they can simply add them). FileMaker provides an excellent graphical user interface (GUI) and a stable hosting environment that allows for simple changes by users to the database architecture and format. If we were to contract a programmer to create a MySQL database from scratch, it would cost several thousand dollars, the GUI would be fixed and modifying it would be a significant hurdle for end users unless they were quite adept at actual coding (which FileMaker does not require). This database is not the point of the paper, it is merely another resource that we offer as a tool to facilitate record keeping and plasmid distribution and has no bearing on the utility of the toolkit.

SUGGESTIONS TO AUTHORS

Specific comments and questions below

The manuscript is long and detailed. Some of the summarisation of the existing gateway/tol2kit could be condensed since this is published and widely used for many years

Agreed and point taken.

The text from p34 moves over into p36 (figure 7 in between), the end of 34 is the end of a sentence but p36 this starts in the middle of a sentence, so I can't tell what is being said.

Apologies and corrected.

p40 destabilised tTA already abbreviated

Thank you.

typos - combing should be combining

Thank you.

NK-kB/p65 should be NF-kB/p65 I think

Exactly and thank you.

FIGURES

Figure 1

I'm not sure this is needed. individual figures throughout the manuscript have this detail as well and the gateway system is well described both by the company and more specific to this work in the

tol2kit wiki page and manuscript. I suggest moving to supplemental

Agreed. Done.

Figure 2

The lifeact fluorophores seem to label fewer cells than phalloidin. Is this expected for this system?

In Figure 2 the cells were transiently transfected, and admittedly there is low efficiency in some of these panels (for instance, in panel B only 2 cells out of 16 in the field of view are expressing the Lifeact plasmid). However, phalloidin staining (which is a dye) will label every cell (while transient transfection will not). We hope this clears up the reason for the difference in signal between the two reagents. At the cellular level, cells that are Lifeact-mScarlet-I positive show clear F-actin stress fiber labeling. We also now include images in Fig. S3 using actin vhh and lifeact and show their overlap.

Figure 3

While this figure demonstrates the utility of the optogenetically controlled Cre, as far as I can tell it does not utilise new components developed in the MAGIC toolkit. Is this the case?

The original paper describing lightCre did not show any in vivo use of the construct and showed validation only in yeast and cultured mammalian cells (not in vivo).

Figure 3 uses new components made in the MAGIC kit, specifically the pCAGEN- LiCre (JDW 1220), which is derived from pME-LiCre (JDW 1119).

We failed to describe pCMV LiCre (JDW 1495) in the original submission. This was generated by recombination of pME-LiCre with a pLenti CMV backbone (Addgene # 17451). We have corrected this omission in the revised manuscript. We thank the Reviewer for pointing out this oversight.

Figure 4

The legend states that 405nm laser is used whereas the methods state 488nm laser. Please clarify

The 488 laser was used. This has been corrected in the manuscript.

Figure 5

In 5G the promoter with the TRE is labelled TREtight and TRE3G in the figure. I think these the same? Only TRE3G is listed in the plasmid table

Thank you for catching this error. The figure has been corrected.

Figure 10

pA is labelled in some but not all constructs

Done.

In C the mODC is labelled as dest (I think) . Not sure why this abbreviation?
in 10D the construct does not look to have the mODC but in the text it does. Please clarify

Apologies, DEST was referring to “destabilized” (the intended consequence of adding the mODC peptide). We recognize how this would cause confusion with DESTINATION in a paper describing Gateway cloning. We have relabeled all figures with mODC.

Table

pME Vhh mCherry is listed as being shown in figure 2. I can't see this in figure 2

We apologize. We had swapped the figure out for the pME Actin Vhh-mNeonGreen and the HALO tag variants instead. We have corrected Table 3 to reflect this change.

Some icre constructs are labelled as being codon optimised - which species are the optimised for?

iCre vectors are codon optimized for mammals (as per Shimshek DR et al., *m Genesis*, 2002; PMID: 11835760), while zCre is codon optimized for zebrafish (using Cre.zf1 from Horststick et al., *Nucleic Acids Research*, 2015).

The intron split zCre design was based on a similar approach that was first validated in mammalian cell culture and later in mice, as referenced in the manuscript. We have added this information to the manuscript to clarify the design for readers.

Some of the tissue specific dest vectors do not say which species their promoter is derived from

Thank you. We have added this into the description and in the Tables.

It might be helpful given the last 2 columns to have a "species" column added to the table

We have attempted to add clarifying columns to all tables. We hope that this addresses their concerns.

Second decision letter

MS ID#: dev.204308R1

MS TITLE: Multisite Assembly of Gateway Induced Clones (MAGIC): a flexible cloning toolbox with diverse applications in vertebrate model systems.

AUTHORS: William Gillespie; Yuwen Zhang; Oscar Ruiz; Juan Cerda III; Joshua Ortiz-Guzman; Michelle Sherman; Williamson Turner; Gabrielle Largoza; Lili Mosser; Esther Fujimoto; Chi-Bin Chien; Kristen Kwan; Benjamin Arenkiel; Walter Patrick Devine; Joshua Wythe

Dear Dr Wythe,

Apologies for the delay in making a decision on your manuscript. We received detailed reviews from two of the three reviewers, which are appended below, or you can access them online: please go to . As you will see, the two reviews support publication of your manuscript, subject to minor revisions. In addition, as concerns were raised about the novelty of some of the reagents and the appropriate citation of the existing literature in your initial submission, we have looked at this issue in your revised manuscript. Below, we indicate where we think that you should consider additional citations and/or clarifications.

- Some of the rows in Tables 5 and 6 are missing an entry for "Lab origin/citation" - where did these vectors originate?
- We suggest that you consider including a seventh table, detailing the existing vectors used to derive the vectors in your study, and cite their origins (e.g. pME-loxP-H2B-mCerulean-2xSTOP-loxP, pME-loxP-H2B-mCerulean-2xSTOP-loxP, etc.). This information is in the materials and methods, but it would be helpful to summarise these existing vectors in a list and would make the contributions of other labs more transparent.
- We suggest that you consider rephrasing this sentence:

"For this reason, we present MAGIC, a collection of novel Entry and Destination vectors to facilitate transgenesis in mammalian systems."

To text along the lines:

"For this reason, we present MAGIC, a collection of Entry and Destination vectors, generated de novo or derived from existing vectors, to facilitate transgenesis in mammalian systems."

There are two specific issues raised by the reviewers on the original submission that we feel need clarification, as detailed below:

Reviewer 2 comment:

3 a) *"p5E-Ubi (JDW 804), which contains a ~3.5 kb insert spanning the promoter, exon1, and intron 1 of the zebrafish ubiquitin locus for constitutive expression in Danio rerio" - this is re-cloning of the zebrafish ubiquitin promoter from a more complex vector into a gateway 5' vector, which seems identical to the Zon lab's widely used ubb 5' vector pCM206 available on AddGene (Mosimann et al., 2011).*

Author response:

In this particular instance we failed to realize a p5E-Ubi vector was already available/derived from the same parental vector that we had used to generate our p5E Ubi plasmid (the substrate being, as detailed and cited in the materials and methods section of the original draft, p5E-Ubi-loxP-EGFP-stop-loxP, Addgene # 27322, from Mosimann et al., 2011). We then validated our re-made version in the hot-Cre system in Figure 10 of the original manuscript. As far as we can tell, while the Addgene plasmid and ours differ by 2 nucleotides, this is likely because our recent long range nanopore sequencing picked up something different than when Dr. Mosimann's p5E-Ubi / Addgene # 27320 was originally deposited and sequenced at Addgene. As the two vectors overlap exactly at the 3' edge of the BamHI site, we believe we repeated exactly what they had done with the same parental vector (thus inadvertently re-making the vector, as despite the 2 bp discrepancy we believe they are the same). We have removed this plasmid from our manuscript and from Addgene. We sincerely thank the Reviewer for pointing out this oversight out to us.

Editor comment:

You acknowledge this in the LR cloning section upon first mention (quoted below), which seems fair enough, but why not also cite Mosimann et al., 2011?

"p5E-Ubi-pro (JDW 804, which we created but is identical to Addgene # #27320)"

Reviewer 2 comment:

3 b) *The SV40 poly A vector (JDW461) sounds like #302 from the original Tol2 kit (Kwan et al., 2007), which is puzzling given the first-author of the original study and late PI are on the manuscript as co-authors.*

c) *Vectors JDW 755, 760, and 761 sound again like vectors already introduced in Kwan et al., 2007, specifically #394 and #395 that are the most commonly used Tol2 backbones in the zebrafish field.*

Author response:

As detailed in Table 4 in the "Origin/Publication" column (far right side of the table), the plasmid is p3E-polyA from Kwan et al., 2007. We have our own plasmid unique ID system in the lab (as shown in the supplemental database we included) to track all cloning/plasmid generation steps. We hoped that listing the plasmid source in the table and citing the relevant manuscript would make clear what we had generated in the paper vs what was derived before (hence the "Wythe lab/this paper", "Kwan lab/this paper" vs "Chien Lab / Kwan et al., Dev Dyn" annotations in the Tables, and also why we don't describe it as being generated in the text of the paper or the materials and methods section).

In the revised manuscript, we have eliminated the internal JDW numbering for all published plasmids (while retaining the far-right side column citing the source of the plasmid). We hope

that this, along with the clear annotations in the tables, make the origin of all referenced plasmids clear to readers.

Editor comment:

From what we understand, you originally gave 'JDW' names to some existing plasmids, which the reviewers interpreted as you creating - or taking credit for - them. We think that in the original paper, the origins of these plasmids were acknowledged in Table 4 but you have now chosen to remove any mention of them completely and only catalogue "novel" vectors. We think this makes it clear they are not part of this new toolkit, but we're unsure why they were mentioned at all in the original paper if they were not created as part of this work and were not used for deriving any of the new vectors - (at least we didn't notice mention of the plasmids labelled JDW 755, 76- and 761 mentioned in the materials and methods). Could you please clarify?

We note that JDW 461 is still mentioned in the LR cloning section (quoted below), but the plasmid name here (p35 v5-mScarlet-1), and the name of the JDW 461 plasmid in the original submission (p3E-polyA) are different, so this could be a mistake? In the revision, the p35 v5-mScarlet-1 plasmid has no reference and there's no indication this was created by the authors since it's not included in the tables. It's possible there are more instances of vectors not in the tables but also not provided with references, so you should check all this carefully, and it would be easy to spot in a table format as suggested above.

pTol2-Hsp70-lox-H2B-mCerulean-stop-lox-mScarlet; fli1ep-zCrel-BFP (JDW 1233) was generated by an LR reaction (p5E-Hsp701) (JDW 458 / Chien Lab #222)(Kwan et al., 2007), pME-loxP-H2B-mCerulean-2xSTOP-loxP (a kind gift of Drs. Michael Harrison and Ellen Lien at USC)(Harrison et al., 2015), p3E V5-mScarlet-1 (JDW 461), and pTol2-Dest-fli1ep-zCrel-BFP (JDW 1218).

In summary, we will be happy to publish your manuscript, subject to addressing the reviewer comments and the points raised above. Please attend to all of the editor/reviewers' comments in your revised manuscript and detail them in your point-by-point response.

Reviewer 1

SUMMARY OF THE ADVANCE MADE IN THIS PAPER AND ITS POTENTIAL SIGNIFICANCE TO THE FIELD

The authors have addressed all of my concerns. I just have some suggestions concerning the text. Please see below.

SUGGESTIONS TO AUTHORS

Abstract: remove "we" in line 9
 Please check line 195-197
 Please check line 228, 230, 231
 Please correct "Hoescht" to "Hoechst"
 Fig.3 J please correct (Tg(kdrl:EGFP))
 Please check line 263, 264, 267
 Please update the scale bars (often not visible, too small)
 Please check line 357
 Line 424-426 please provide references

Reviewer 2

No comments to add.

Reviewer 3

SUMMARY OF THE ADVANCE MADE IN THIS PAPER AND ITS POTENTIAL SIGNIFICANCE TO THE FIELD

the revised version has addressed all of my comments. The structure and length are improved and the details of the plasmids clearer in the revised tables

Second revision

Author response to reviewers' comments

Editorial Comments

Some of the rows in Tables 5 and 6 are missing an entry for "Lab origin/citation" - where did these vectors originate?

Apologies, those were added late to the revision. They originated in our laboratory. Those fields are now complete in the revised Tables.

- We suggest that you consider including a seventh table, detailing the existing vectors used to derive the vectors in your study, and cite their origins (e.g. pME-loxP-H2B-mCerulean-2xSTOP-loxP, pME-loxP-H2B-mCerulean-2xSTOP-loxP, etc.). This information is in the materials and methods, but it would be helpful to summarise these existing vectors in a list and would make the contributions of other labs more transparent.

As the materials and methods clearly state this information, in extensive detail, this would be redundant and time consuming, as well as complicated, as the origin often entails multiple pieces that were joined together and lacks context, therefore we have chosen not to include this additional Table for the >150 vectors.

- We suggest that you consider rephrasing this sentence:

"For this reason, we present MAGIC, a collection of novel Entry and Destination vectors to facilitate transgenesis in mammalian systems."

To text along the lines:

"For this reason, we present MAGIC, a collection of Entry and Destination vectors, generated de novo or derived from existing vectors, to facilitate transgenesis in mammalian systems."

Done.

There are two specific issues raised by the reviewers on the original submission that we feel need clarification, as detailed below:

Reviewer 2 comment:

3 a) "p5E-Ubi (JDW 804), which contains a ~3.5 kb insert spanning the promoter, exon1, and intron 1 of the zebrafish ubiquitin locus for constitutive expression in Danio rerio" - this is re-cloning of the zebrafish ubiquitin promoter from a more complex vector into a gateway 5' vector, which seems identical to the Zon lab's widely used ubb 5' vector pCM206 available on AddGene (Mosimann et al., 2011).

Author response:

In this particular instance we failed to realize a p5E-Ubi vector was already available/derived from the same parental vector that we had used to generate our p5E Ubi plasmid (the substrate being, as detailed and cited in the materials and methods section of the original draft, p5E-Ubi-loxP-EGFP-stop-loxP, Addgene # 27322, from Mosimann et al., 2011). We then validated our re-

made version in the hot-Cre system in Figure 10 of the original manuscript. As far as we can tell, while the Addgene plasmid and ours differ by 2 nucleotides, this is likely because our recent long range nanopore sequencing picked up something different than when Dr. Mosimann's p5E-Ubi / Addgene # 27320 was originally deposited and sequenced at Addgene. As the two vectors overlap exactly at the 3' edge of the BamHI site, we believe we repeated exactly what they had done with the same parental vector (thus inadvertently re-making the vector, as despite the 2 bp discrepancy we believe they are the same). We have removed this plasmid from our manuscript and from Addgene. We sincerely thank the Reviewer for pointing out this oversight out to us.

Editor comment:

You acknowledge this in the LR cloning section upon first mention (quoted below), which seems fair enough, but why not also cite Mosimann et al., 2011?

"p5E-Ubi-pro (JDW 804, which we created but is identical to Addgene # #27320)"

In the original submission when we described how we made the p5E-Ubi vector (that lacked the lox-stop-lox-EGFP insert) we did cite their paper when referring to the parental vector, as shown in the screen shots below and can be found in the original PDF we submitted to *Development* and in our preprint.

p5E Ubi-pro, which contains the zebrafish *Ubiquitin* promoter, exon1, and intron 1, was made by digesting p5E-Ubi-loxP-EGFP-loxP (Addgene #27322)¹⁵³ with BamHI, and ligating the digested product together to remove the loxP flanked EGFP cassette and reconstitute the BamHI site.

¹⁵³ Mosimann, C., Kaufman, C. K., Li, P., Pugach, E. K., Tamplin, O. J. & Zon, L. I. Ubiquitous transgene expression and Cre-based recombination driven by the ubiquitin promoter in zebrafish. *Development* **138**, 169-177, doi:10.1242/dev.059345 (2011).

As previously explained in the rebuttal, we independently generated p5E-Ubi from the parental plasmid (p5E-Ubi-lox-stop-lox-EGFP, Addgene #27322), which we obtained from Addgene (as cited in the original paper, shown and explained above). We clearly identified the provenance of the parental p5E-Ubi-lox-stop-lox vector, providing the Addgene source/record and citing the paper in the materials and methods of our manuscript (thus we were clearly not attempting to obfuscate the record, if that is what the Editor is suggesting). I am not clear what more we were supposed to do when we were not aware, as explained in the rebuttal, of the p5E-Ubi-pro plasmid at Addgene (lacking the lox-stop-lox, #27320) and as soon as we became aware of this oversight we removed our independently generated p5E-Ubi-pro plasmid from Addgene and from the paper).

Reviewer 2 comment:

3 b) The SV40 poly A vector (JDW461) sounds like #302 from the original Tol2 kit (Kwan et al., 2007), which is puzzling given the first-author of the original study and late PI are on the manuscript as co-authors.

c) Vectors JDW 755, 760, and 761 sound again like vectors already introduced in Kwan et al., 2007, specifically #394 and #395 that are the most commonly used Tol2 backbones in the zebrafish field.

Author response:

As detailed in Table 4 in the "Origin/Publication" column (far right side of the table), the plasmid is p3E-polyA from Kwan et al., 2007. We have our own plasmid unique ID system in the lab (as shown in the supplemental database we included) to track all cloning/plasmid generation steps. We hoped that listing the plasmid source in the table and citing the relevant manuscript would make clear what we had generated in the paper vs what was derived before (hence the "Wythe lab/this paper", "Kwan lab/this paper" vs "Chien Lab / Kwan et al., Dev Dyn" annotations in the Tables, and also why we don't describe it as being generated in the text of the paper or the

materials and methods section).

In the revised manuscript, we have eliminated the internal JDW numbering for all published plasmids (while retaining the far-right side column citing the source of the plasmid). We hope that this, along with the clear annotations in the tables, make the origin of all referenced plasmids clear to readers.

Editor comment:

From what we understand, you originally gave 'JDW' names to some existing plasmids, which the reviewers interpreted as you creating - or taking credit for - them.

Exactly.

We think that in the original paper, the origins of these plasmids were acknowledged in Table 4

Yes, they were (as clearly discussed in our rebuttal).

but you have now chosen to remove any mention of them completely and only catalogue "novel" vectors.

Given the Reviewer's confusion and tone of their comments we considered removing these plasmids entirely from the Tables the easiest way to avoid alternative interpretations, as even when we provided the relevant sources in the original manuscript in Tables, and in the materials and methods, our intent was still misinterpreted.

Thus, in the revised manuscript we only discuss alternative publicly available plasmids in the main text and do not list them in the Tables (even if they were used in a multisite Gateway reaction, as one of four plasmids, to create a final expression clone, they are not listed in the Tables but are cited in the materials and methods).

We think this makes it clear they are not part of this new toolkit, but we're unsure why they were mentioned at all in the original paper if they were not created as part of this work and were not used for deriving any of the new vectors - (at least we didn't notice mention of the plasmids labelled JDW 755, 76- and 761 mentioned in the materials and methods). Could you please clarify?

As explained in the rebuttal, assigning a JDW # to every plasmid in our lab (which numbers over 1500 at this point) is the first step in our internal laboratory information management system (as evident from the supplied FileMaker database we provide). Some plasmids were listed for comparison's sake, so readers could know what other options were publicly available at Addgene for such common things as a CMV promoter clone, or a p3E polyA clone, for instance, and how ours new plasmids filled a new niche (such as our pCAG promoter). Again, the origin and manuscripts for all plasmids were clearly cited in the Tables.

JDW 761/ pTol2DestCG2, as explained in the materials and methods, was used to generate the p5E-myh7 promoter clone, and thus included there, but also shown in the Table as an example pTol2 DEST plasmid as this is a major workhorse of the field, and was also used as a backbone for some of the constructs in the paper (and the lead author of that manuscript, Dr. Kwan, is on this paper, as is the corresponding author, Dr. Chien—albeit it posthumously).

Multisite Promoterless DEST Tol2 Vectors	JDW #	Description	Uses	Figures	Origin
pDestTol2pA*	JDW 755	attR4-R3 gate with SV40 polyA, flanked by Tol2 inverted repeats	Multisite Tol2 Cloning, zebrafish transgenesis		Chien Lab, Kwan et al., Dev Dyn.
pDestTol2pA2	JDW 760	pDestTol2pA with ~2 kb extraneous sequence removed	Multisite Tol2 Cloning, zebrafish transgenesis		Chien Lab, Kwan et al., Dev Dyn.
pDestTol2CG2	JDW 761	pDestTol2pA with cardiomyocyte-specific GFP reporter in opposite direction	Multisite Tol2 Cloning, zebrafish transgenesis		Chien Lab, Kwan et al., Dev Dyn.

JDW 760 / pDEST Tol2pA2 was used as a template for subcloning, as explained in the original

manuscript (screen shot below) and again, provided as an example of an available Tol2 backbone (while pDEST Tol2pA*, a truncated version, was shown for comparison, as this is a major workhorse in the field and we felt provided context for how our new plasmids fit in/compared/contrasted with available tools).

pDestTol2-2xIns; hsp70-zCrel-BFP-mODC (JDW 1002) was digested with Sall and Cold Fusion (SBI Bioscience) was done to insert a Gateway recombination cassette upstream of the polyA using the primers and JDW 760 (pTol2Dest-pA2) as a template with the following primers:

We note that JDW 461 is still mentioned in the LR cloning section (quoted below), but the plasmid name here (p35 v5-mScarlet-1), and the name of the JDW 461 plasmid in the original submission (p3E-polyA) are different, so this could be a mistake? In the revision, the p35 v5-mScarlet-1 plasmid has no reference and there's no indication this was created by the authors since it's not included in the tables. It's possible there are more instances of vectors not in the tables but also not provided with references, so you should check all this carefully, and it would be easy to spot in a table format as suggested above.

pTol2-Hsp70-lox-H2B-mCerulean-st op-lox-mScarlet; fli1ep-zCrel-BFP (JDW 1233) was generated by an LR reaction (p5E-Hsp701) (JDW 458 / Chien Lab #222)(Kwan et al., 2007), pME-loxP-H2B-mCerulean-2xSTOP-loxP (a kind gift of Drs. Michael Harrison and Ellen Lien at USC)(Harrison et al., 2015), p3E V5-mScarlet-1 (JDW 461), and pTol2-Dest-fli1ep-zCrel-BFP (JDW 1218).

p3E V5 mScarlet is JDW 968 and was included in the original and revised manuscript tables (and used to create this expression plasmid mentioned above, JDW 1233). This error in the text been assigning JDW 461 to that plasmid has been corrected.

The numbering for the previously published plasmids has been removed and they are referenced and an Addgene # provided (when applicable). We have verified the numbering is correct (and is retained at Addgene, so these tools can be easily tracked and obtained by other investigators).

Comments from the Reviewers:

Reviewer 1:

The authors have addressed all of my concerns. I just have some suggestions concerning the text. Please see below.

We thank the Reviewer for their detailed comments and have addressed all of these comments

SUGGESTIONS TO AUTHORS

Abstract: remove "we" in line 9

Done (but not sure why this was necessary to change other than stylistic preference).

Please check line 195-197

Thank you, corrected.

Please check line 228, 230, 231

The JDW reference #s have been removed.

Please correct "Hoescht" to "Hoechst"

We cannot find Hoescht in the manuscript text, only Hoechst (line 1625)

Fig.3 J please correct (Tg(kdrl:EGFP))

Done.

Please check line 263, 264, 267

Done

Please update the scale bars (often not visible, too small)

Done

Please check line 357

Done

Line 424-426 please provide references

The statement that our newer vectors have improved kinetics compared to the original pB-Tet-On is not historical, thus references do not apply, but are instead illustrated by Figure 7B and C, which show less basal activity in pB-Tet-On 2.0 and 3.0 than pB-Tet-On (thus it is “tighter” and less promiscuous) in absence of Dox (Figure 7B). Moreover, 2.0 shows comparable activation/induction to pB-Tet-On following Dox treatment, with tighter regulation (see the error bars/spread of data in Figure 7C post Dox). Finally, Figure 7D shows that induction of pB-Tet-On 2.0 is robustly downregulated within 24 hours post Dox removal. Overall, these data show the newer variants display improved on/off kinetics compared to the original pB-Tet-On.

Reviewer 2:

No comments to add.

Reviewer 3:

the revised version has addressed all of my comments. The structure and length are improved and the details of the plasmids clearer in the revised tables

Third decision letter

MS ID#: dev.204308R2

MS TITLE: Multisite Assembly of Gateway Induced Clones (MAGIC): a flexible cloning toolbox with diverse applications in vertebrate model systems.

AUTHORS: William Gillespie; Yuwen Zhang; Oscar Ruiz; Juan Cerda III; Joshua Ortiz-Guzman; Michelle Sherman; Williamson Turner; Gabrielle Largoza; Lili Mosser; Esther Fujimoto; Chi-Bin Chien; Kristen Kwan; Benjamin Arenkiel; Walter Patrick Devine; Joshua Wythe

ARTICLE TYPE: Techniques and Resources Article

Dear Joshua,

Apologies for the delay in looking over your revisions but happy to tell you that your manuscript has now been accepted for publication in Development, pending our standard publication integrity checks.